# IDH1 regulates human erythropoiesis by eliciting chromatin state reprogramming

Mengjia Li[1,2†], Hengchao Zhang[1†], Xiuyun Wu[1†], Mengqi Yu[1], Qianqian Yang[1], Lei Sun[1], Wei Li[2], Zhongxing Jiang[2], Fumin Xue[3], Ting Wang[1]*, Xiuli An[4]*, Lixiang Chen[1]*

[1]State Key Laboratory of Metabolic Dysregulation and Prevention and Treatment of Esophageal Cancer; School of Life Sciences, Zhengzhou University, Zhengzhou, China; [2]Department of Hematology, First Affiliated Hospital of Zhengzhou University, Zhengzhou, China; [3]Department of Gastroenterology, Children's Hospital Affiliated to Zhengzhou University, Zhengzhou, China; [4]Laboratory of Membrane Biology, New York Blood Center, New York, United States

*For correspondence:
tingwang@zzu.edu.cn (TW);
xan@nybc.org (XA);
lxchen@zzu.edu.cn (LC)

[†]These authors contributed equally to this work

Competing interest: The authors declare that no competing interests exist.

## eLife Assessment

This study by Li et al. presents **important** findings on the metabolism-independent role of nuclear IDH1 in chromatin regulation during erythropoiesis. The authors provide **convincing** evidence that IDH1 deficiency disrupts H3K79 methylation and nuclear architecture, contributing to dyserythropoiesis. Their findings offer invaluable mechanistic insights with potential therapeutic implications for erythroid disorders and hematologic malignancies.

**Abstract** Isocitrate dehydrogenase 1 (IDH1) is the key enzyme that can modulate cellular metabolism, epigenetic modification, and redox homeostasis. Gain-of-function mutations and decreased expression of IDH1 have been demonstrated to be associated with pathogenesis of various myeloid malignancies characterized by ineffective erythropoiesis, such as acute myeloid leukemia (AML) and myelodysplastic syndrome (MDS). However, the function and mechanism of IDH1 in human erythropoiesis still remains unclear. Here, utilizing the human erythropoiesis system, we present an evidence of IDH1-mediated chromatin state reprogramming besides its well-characterized metabolism effects. We found that knockdown IDH1 induced chromatin reorganization and subsequently led to abnormalities biological events in erythroid precursors, which could not be rescued by addition of reactive oxygen species (ROS) scavengers or supplementation of α-ketoglutarate (α-KG). We further revealed that knockdown IDH1 induces genome-wide changes in distribution and intensity of multiple histone marks, among which H3K79me3 was identified as a critical factor in chromatin state reprogramming. Integrated analysis of ChIP-seq, ATAC-seq, and RNA-seq recognized that SIRT1 was the key gene affected by IDH1 deficiency. Thus, our current work provided novel insights for further clarifying fundamental biological function of IDH1 which has substantial implications for an in-depth understanding of pathogenesis of diseases with IDH1 dysfunction and accordingly development of therapeutic strategies.

## Introduction

IDH1 is a key enzyme involved in regulation of redox homeostasis by catalyzing oxidative decarboxylation of isocitrate to α-KG, also known as 2-oxoglutarate (2-OG), simultaneously reducing $NAD(P)^+$ to NAD(P)H and liberating CO (**Waitkus et al., 2018**). Growing evidence suggests that, in addition to metabolism regulation, IDH1 is also involved in regulation of epigenetic landscape (**Chan et al.,**

*2015*). IDH1 mutations are known to cause dysregulation of chromatin modifications, such as DNA methylation, histone acetylation and methylation, which are associated with the occurrence and development of various myeloid malignancies characterized by ineffective erythropoiesis, such as AML and MDS (*Ward et al., 2010*; *Xu et al., 2011*). Notably, recent research work also revealed that decreased expression of IDH1 are also contributed to the onset of AML and MDS (*Figueroa et al., 2010*; *Rohle et al., 2013*).

Previous studies found that IDH1 mutations led to loss of normal catalytic activity (*Tateishi et al., 2015*) and gain of function, causing accumulation of the rare metabolite 2-hydroxyglutarate (2-HG) (*Turcan et al., 2018*; *Su et al., 2018*). 2-HG acts as an 'oncometabolite' by competitively inhibiting multiple α-KG-dependent dioxygenases (*Ward et al., 2010*; *Xu et al., 2011*; *Figueroa et al., 2010*; *Rohle et al., 2013*; *Tateishi et al., 2015*; *Turcan et al., 2018*; *Su et al., 2018*; *Chen et al., 2009*; *Baron and Barminko, 2016*; *Mei et al., 2021*; *Gonzalez-Menendez et al., 2021*; *Gu et al., 2021*; *Chung et al., 2020*; *Liu et al., 2019*; *Trakarnsanga et al., 2017*; *Karahoda et al., 2022*; *Zheng et al., 2020*; *Sun et al., 2022*; *Grewal and Moazed, 2003*; *Goasguen et al., 2018*; *Strahl and Allis, 2000*; *Tan et al., 2011*; *Shah and Henriksen, 2011*; *Thomas et al., 2022*; *Gut and Verdin, 2013*; *Yego and Mohr, 2010*; *Zhang et al., 2019*; *Mingay et al., 2018*; *Palis, 2008*; *Hattangadi et al., 2014*; *Godfrey et al., 2021*; *Godfrey et al., 2019*; *Li et al., 2018*; *Kuno et al., 2022*; *Liu and Shi, 2022*; *Zhang et al., 2023*; *Rajendran et al., 2011*; *Dai et al., 2017*; *Li et al., 2014*; *Huang et al., 2019*; *Pollyea et al., 2021*; *Stahl et al., 2023*; *Heckl et al., 2014*; *Wen et al., 1995*; *Morgenstern and Land, 1990*; *Qu et al., 2018*; *Li et al., 2023*; *Kim et al., 2015*; *Anders et al., 2015*; *Anders and Huber, 2010*), consequently altering cells epigenetic state and leading to pathogenesis of myeloid malignancies (*Waitkus et al., 2018*; *Su et al., 2018*). However, several recent studies have shown that suppression of mutant IDH1 expression and/or inhibition of 2-HG production is not sufficient to reverse mutant IDH1-induced epigenetic changes (*Rohle et al., 2013*; *Tateishi et al., 2015*; *Turcan et al., 2018*). Blocking production of 2-HG did not inhibit the growth of many solid cancers with IDH1 mutations (*Tateishi et al., 2015*). Importantly, although IDH1 mutations lead to gradual accumulation of DNA and histone methylation markers, inhibition of IDH1 mutation expression did not lead to complete restoration of epigenome and transcriptome to initial state (*Turcan et al., 2018*). Moreover, it has been reported that high concentration of 2-HG produced by mutant IDH1 also has antineoplastic activity in glioma (*Su et al., 2018*). All these findings strongly suggested that the roles of IDH1 mutations in the pathogenesis of AML and MDS might not only be attributed to its capability to produce 2-HG.

Considering that genetic mutations may lead to the loss of original functions while acquiring new roles, we speculate that the roles of IDH1 mutations in the pathogenesis of myeloid disorder diseases might not only be attributed to the gain of function-dependent manner, but also attributed to loss of function manner, which has not been recognized previously. It also should be noted that most previous studies focused primarily on the pathogenic roles of mutated IDH1, whereas the biological function of IDH1 and underlying mechanisms remain largely unknown.

In this study, by using an in vitro human erythropoiesis system, during which cell morphology and chromatin architecture are dramatically altered (*Chen et al., 2009*; *Baron and Barminko, 2016*), we presented an evidence that nuclear IDH1 is involved in chromatin state reprogramming to regulate terminal erythropoiesis.

## Results

### IDH1 deficiency impaired terminal erythropoiesis

Erythropoiesis is a continuous and complex process, it can be divided into two stages: early erythroid development and terminal erythroid differentiation. To further explore the roles of IDH1 during human erythropoiesis, we first analyzed the expression level of IDH1 from the transcriptomics data of highly purified populations of erythroid cells from cord blood at distinct stages of erythropoiesis. As shown in *Figure 1—figure supplement 1A*, the expression of IDH1 was low in hematopoietic stem cells, but high in erythroid progenitor cells and late erythrocytes erythropoiesis. We collected cells on days 7, 9, 11, 13, and 15 during the process of erythroid differentiation and detected the mRNA and protein level of IDH1. As shown in *Figure 1—figure supplement 1B, C and D*, IDH1 was consistently expressed at all stages of erythroid development. Then, we performed shRNA/siRNA-mediated knockdown of IDH1 on days 7, 11, and 15. ShRNA and siRNA-mediated knockdown efficiency were

confirmed by qRT-PCR (*Figure 1—figure supplement 2A*) and western blot (*Figure 1—figure supplements 2B, C and 3A, B*). Consistent with our previous results, we found that loss of IDH1 had a slight effect on cell growth (*Figure 1—figure supplement 2D*) but not on apoptosis at late stages of erythropoiesis (*Figure 1—figure supplements 2E, F and 3C, D*). Although the differentiation of colony-forming unit-erythroid (CFU-E) cells into terminal stage, characterized by glycoprotein A expression, was not affected by lack of IDH1 (*Figure 1—figure supplements 3E, F and 4A, B*), the generation of orthochromatic cells, defined as GPA$^{pos}$α4-integrin$^{low}$band3$^{hi}$ [12], was delayed by IDH1 deficiency (*Figure 1—figure supplements 4C, D and 5A, B*). This delay of differentiation was also confirmed by morphological observation using cytospin assay (*Figure 1—figure supplement 4E*). Since chromatin condensation and enucleation are distinguished cellular events of polychromatic and orthochromatic cells (*Mei et al., 2021*), we also evaluated cell morphology and enucleation. Based on the cytospins, both binucleated and multi-nucleated cells were both regarded as a nuclear malformation. There were at least 300 cells in each group, of which the number of abnormal nuclei cells as a percentage of the total number of cells was the proportion of aberrant nuclei. We observed a remarkable accumulation of polychromatic and orthochromatic erythroblasts with abnormal nuclei in IDH1-deficient erythroblasts (*Figure 1A*; *Figure 1—figure supplement 5C, D*) and severely impaired enucleation (*Figure 1B*; *Figure 1—figure supplement 5E, F*). In addition, we found that IDH1 deletion resulted in distinctly larger nuclei compared to controls, with the ratio of nucleus/plasma increased more than twofold in IDH1-deficient erythroid cells (*Figure 1—figure supplement 6A, B*), indicating that IDH1 knockdown impaired nuclear condensation.

## IDH1 localizes to nucleus of human erythroid cells

IDH1 has been widely studied as a key metabolic enzyme localized in cytoplasm or peroxisome. In recent years, it has also been reported that IDH1 localized in nucleoplasm and chromatin of embryonic stem cells (ESCs) (*Liu et al., 2019*). However, the subcellular localization of IDH1 during human erythropoiesis still remain largely unknown. To address this, we systematically analyzed the subcellular location of IDH1 in erythroid cells at different time points on umbilical cord blood-derived CD34$^{+}$ cells induced to normal human terminal erythroid development. Immunofluorescence analysis revealed that IDH1 was localized both in the nucleus and cytoplasm as cells matured (*Figure 2A and B*). We also examined location of IDH1 in human umbilical cord blood-derived erythroid progenitor 2 (HUDEP-2), K562 and HEL cell lines and found that IDH1was also localized both in the nucleus and cytoplasm (*Figure 2C*, *Figure 2—figure supplement 1A, B*). In addition, we obtained paraffin-embedded bone marrow samples from 10 patients with IDH1-mutant AML/MDS, including 4 MDS and 6 AML samples (*Supplementary file 1*). IDH1 mutation sites were reported in previous studies, including R132C, R132G, and R132H *Chan et al., 2015*. We further detected localization of IDH1 by immunofluorescence staining assay, IDH1 was also expressed in both nucleus and cytoplasm of erythroblasts characterized as GPA positive expression (*Figure 2D*, *Figure 2—figure supplement 1C*). In contrast, IDH1 was exclusively localized in cytoplasm of 293T cell line (*Figure 2E*). Western blot analyses further confirmed nuclear location of IDH1 in erythroblast but not in 293T cells (*Figure 2F*). The unique subcellular localization pattern of IDH1 in erythroid cells suggests that it may play a role in erythropoiesis through non-canonical functions.

## IDH1 maintain nuclear morphological on terminal erythropoiesis

Previously, presence of IDH1 in nucleus has been mentioned (*Liu et al., 2019*), but it has not been specifically proven that the function of IDH1 relies on its nuclear localization. To further test the function of IDH1 in maintaining nucleus morphology, we knocked out IDH1 in nucleus while retaining cytoplasmic IDH1 (*Figure 3A*). Firstly, as shown in *Figure 3B*, we knocked out IDH1 using CRISPR-Cas9 technology by infecting with viral particles carrying small guide RNA (sgRNA) sequences targeting IDH1 and sorted with GFP to get Sg-IDH1 HUDEP2 cells, in which IDH1 was totally depleted. Cells overexpressing nuclear export signal (NES)-IDH1 were then transduced into Sg-IDH1 HUDEP2 to deplete nuclear IDH1 while preserving IDH1 in cytoplasm in HUDEP2 cells. After expansion and differentiation (*Trakarnsanga et al., 2017*), we further confirmed the selective localization of IDH1 to nucleus by confocal microscopy. As shown in *Figure 3C and D*, there is no presence of IDH1 in nucleus while the cytoplasm localization of IDH1 was still retaining. We also detected the cell growth of control,Sg-IDH1, Sg-NES-IDH1 and Sg-PLVX-IDH1, there were no significant difference between

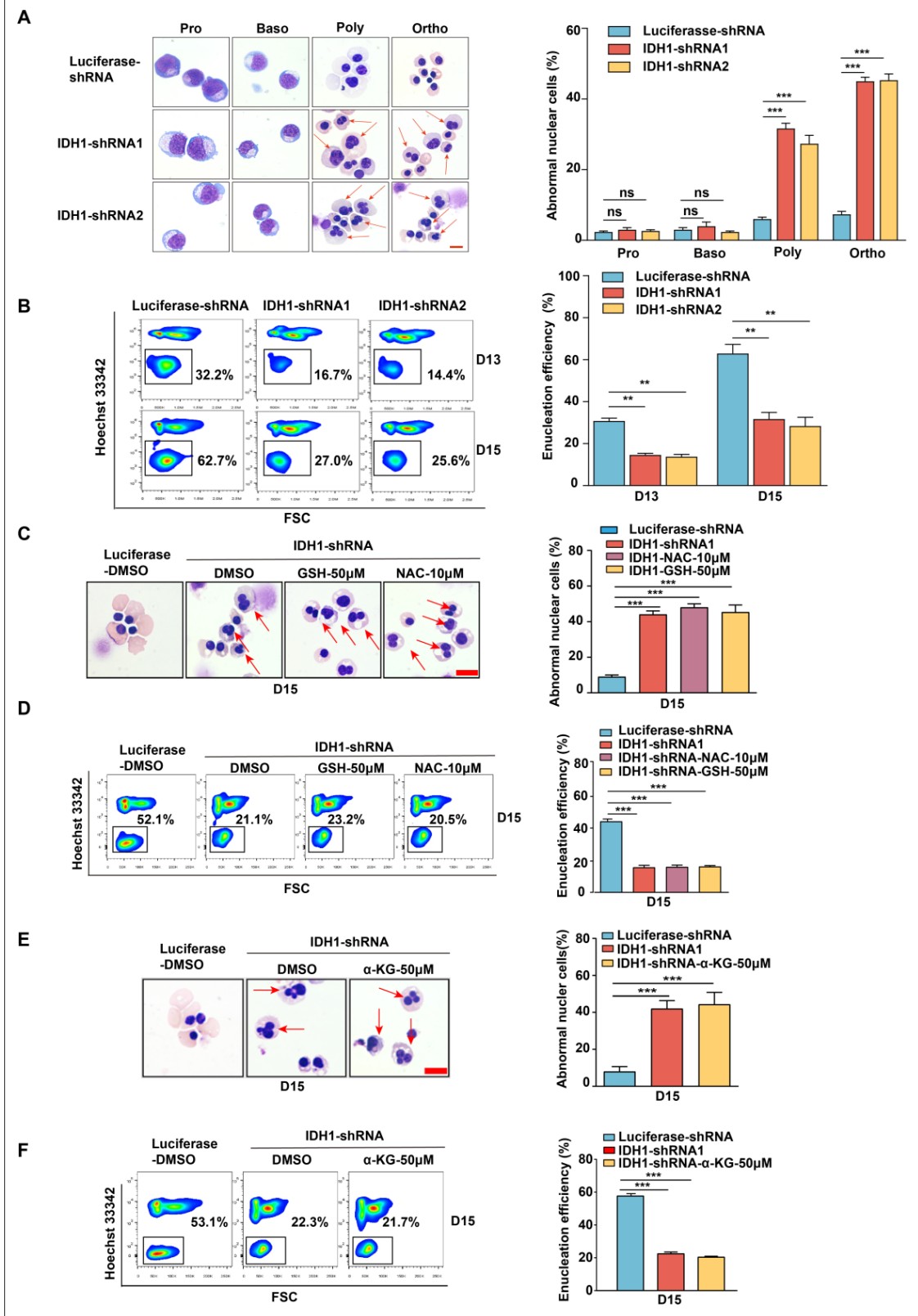

**Figure 1.** Isocitrate dehydrogenase 1 (IDH1) deficiency impaired terminal stage erythropoiesis. (**A**) Representative cytospin images on proerythroblasts, basophilic erythroblasts, polychromatic erythroblasts, and orthochromatic erythroblasts differentiated from cord blood hematopoietic stem cells. The red arrows point to the cells that are abnormal nucleus. Scale bar, 10 µm. Quantitative analysis of the percentage of abnormal nuclear cells from three independent biological experiments. (**B**) Flow cytometry analysis showed the efficiency of enucleation on day 13 and day 15. Quantitative analysis of

*Figure 1 continued on next page*

*Figure 1 continued*

enucleation efficiency from three independent biological experiments. (**C**) Representative cytospin images of erythroblasts after adding glutathione (GSH) (50 µM) and N-acetyl-L-cysteine (NAC) (10 µM) on day 15. Scale bar, 10 µm. Quantitative analysis of the percentage of the cells with abnormal nucleus. (**D**) Flow cytometry analysis showed the enucleation efficiency after adding GSH (50 µM) and NAC (10 µM) on day 15. Quantitative analysis of the enucleation efficiency after adding GSH (50 µM) and NAC (10 µM) on day 15 from three independent biological experiments. (**E**) Representative cytospin images of erythroblasts after supplement α-ketoglutarate (α-KG) (50 µM) on day 15. Scale bar, 10 µm. Quantitative analysis of the percentage of the abnormal nucleus from three independent biological experiments. (**F**) Flow cytometry analysis showed the efficiency of enucleation after supplement α-KG (50 µM) on day 15. Quantitative analysis of the enucleation efficiency after supplement α-KG (50 µM) on day 15 from three independent experiments. Statistical analysis is from three independent biological experiments, and the bar plot represents mean ± SD of triplicate samples. Not significant (ns), *p<0.05, **p<0.01, ***p<0.001.

The online version of this article includes the following source data and figure supplement(s) for figure 1:

**Figure supplement 1.** The expression level of Isocitrate dehydrogenase 1 (IDH1) during erythropoiesis.

**Figure supplement 1—source data 1.** Original western blots for *Figure 1—figure supplement 1C*, indicating the relevant bands.

**Figure supplement 1—source data 2.** Original files for western blot analysis displayed in *Figure 1—figure supplement 1C*.

**Figure supplement 2.** Deficiency of Isocitrate dehydrogenase 1 (IDH1) slightly affect the proliferation and have on effect on apoptosis on terminal erythroblast.

**Figure supplement 2—source data 1.** Original western blots for *Figure 1—figure supplement 2B*, indicating the relevant bands.

**Figure supplement 2—source data 2.** Original files for western blot analysis displayed in *Figure 1—figure supplement 2B*.

**Figure supplement 3.** siRNA-mediated knockdown of isocitrate dehydrogenase 1 (IDH1) have no effect on apoptosis and GPA expression.

**Figure supplement 3—source data 1.** Original western blots for *Figure 1—figure supplement 3A*, indicating the relevant bands.

**Figure supplement 3—source data 2.** Original files for western blot analysis displayed in *Figure 1—figure supplement 3A*.

**Figure supplement 4.** Deficiency of isocitrate dehydrogenase 1 (IDH1) affect the generation of orthochromatic erythroblasts.

**Figure supplement 5.** siRNA-mediated knockdown of isocitrate dehydrogenase 1 (IDH1) impaired the terminal stage erythropoiesis.

**Figure supplement 6.** Deficiency of isocitrate dehydrogenase 1 (IDH1) impaired nuclear condensation.

**Figure supplement 7.** Isocitrate dehydrogenase 1 (IDH1) deficiency induced increase of reactive oxygen species (ROS) and decrease of α-ketoglutarate (α-KG).

groups (*Figure 3—figure supplement 1A*).Then we detected the concentration of α-KG, the results showed that knockout of IDH1 decreased the concentration of α-KG (*Figure 3—figure supplement 1B*). Then we further double staining of Annexin V and 7AAD to detect apoptosis, we found that there were no apoptosis on Sg-IDH1, Sg-NES-IDH1, and Sg-PLVX-IDH1 groups (*Figure 3—figure supplement 1C and D*). We also performed Edu staining to detect cell cycle by FACS, as shown in *Figure 3—figure supplement 1E and F*, we found that cell cycle arrest in G0/G1 and S phase in Sg-IDH1 and Sg-NES-IDH1 groups. Then, we further performed Edu staining to detect cell cycle by FACS, as shown in *Figure 3E and F*, we found that cell cycle arrest in G0/G1 and S phase in Sg-IDH1 and Sg-NES-IDH1 groups. Selective knockdown of nuclear IDH1 caused a significant increase of the proportion of HUDEP2 cells with abnormal nuclei (*Figure 3G*). While, as it was shown in *Figure 3H*, statistical analysis showed that proportion of cell nuclear malformations of Sg-NES-IDH1 cells was as similar as Sg-IDH1 cells. Taken together, these results demonstrated that nuclear IDH1 also plays critical roles in maintaining nuclear morphology during terminal erythropoiesis.

## Deficiency of IDH1 reshaped chromatin landscape in late-stage erythroblasts

Given that nuclear-located chromatin-binding proteins play crucial roles in maintaining chromatin structure and gene expression (*Karahoda et al., 2022*), we further checked the distribution of IDH1 in nucleus of erythroid cells cultivated on day 15. We found that IDH1 is detected in chromatin fraction of erythroid cells on day 15 (*Figure 4A*). To explore the role of IDH1 in maintaining chromatin architecture, we evaluated the effect of IDH1 on the chromatin structure of erythroid cells using transmission electron microscopy. As expected, IDH1 deficiency led to aberrant dynamic transition between euchromatin and heterochromatin state. In mixed late erythroblasts on day 15 as well as purified polychromatic and orthochromatic cells, the proportion of euchromatin increased by 2–3 folds in IDH1-deficient cells compared to control cells (*Figure 4B and C*). There is growing evidence that metabolites drive chromatin dynamics through post-translational modifications (PTMs) that alter

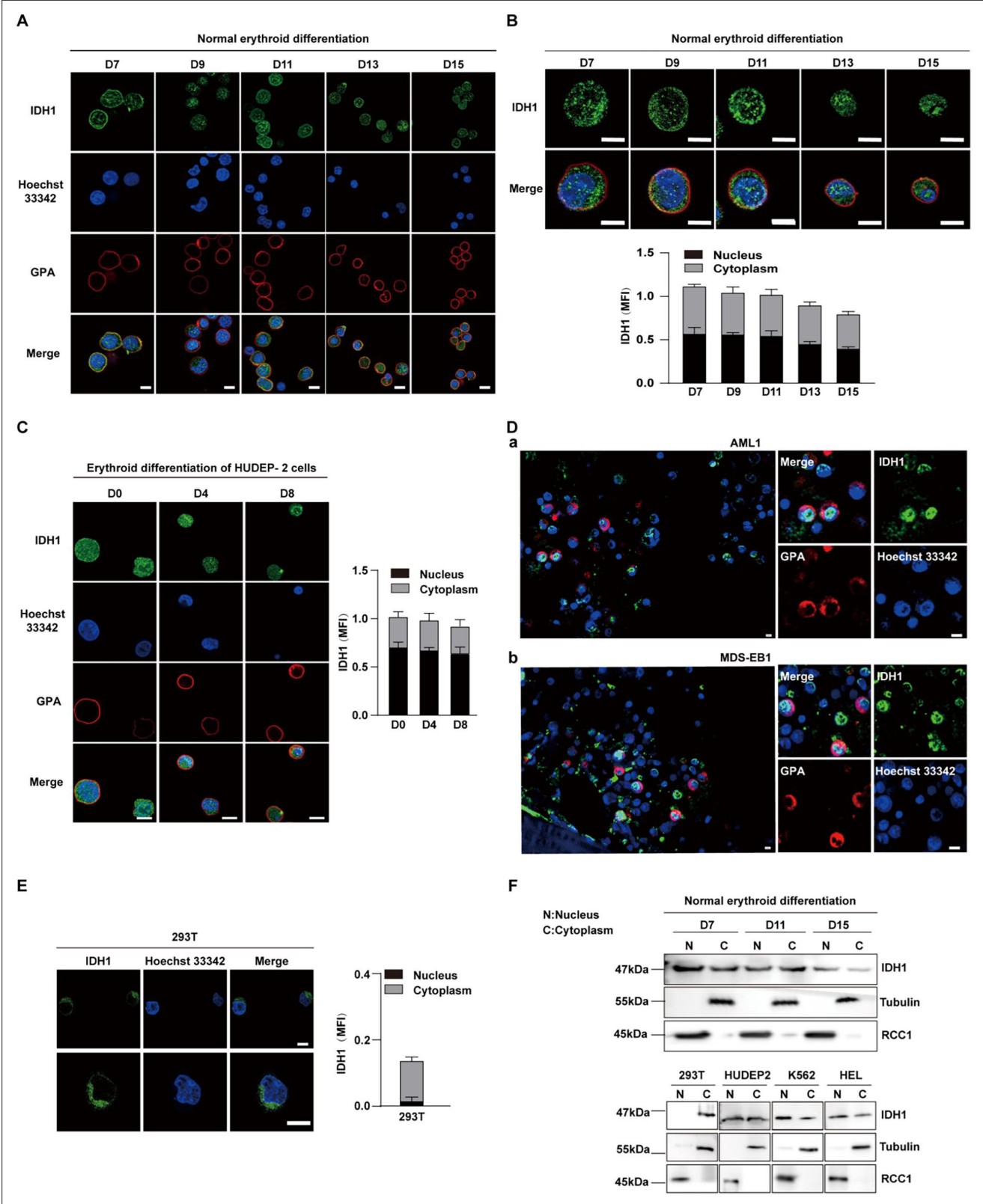

**Figure 2.** Isocitrate dehydrogenase 1 (IDH1) localizes to nucleus in human erythroid cells. (**A, B**) Nuclear location of IDH1 (green) on the terminal stages of erythroid cells. Normal human terminal erythroid cells induced from umbilical cord blood-derived CD34⁺ cells were stained with antibodies targeting IDH1 together with GPA (red) and Hoechst 33342 (blue). Scale bars, 5 μm. MFI (mean fluorescent intensity) of IDH1 in nucleus and cytoplasm during the terminal stages of erythropoiesis was shown at the lower panel. (**C**) Representative immunofluorescence images showed the location of IDH1

*Figure 2 continued on next page*

*Figure 2 continued*

at different time points in the human umbilical cord blood-derived erythroid progenitor 2 (HUDEP-2) cell line. IDH1 (green), GPA (red), and Hoechst 33342 (blue). Scale bars, 5 µm. MFI of IDH1 in nucleus and cytoplasm of erythroid cells was shown at right panel. Data are presented as the mean ± SD from three independent experiments containing at least 30 cells each. (**D**) Representative immunofluorescence images of IDH1 (green), GPA (red), and Hoechst 33342 (blue) staining of the paraffin-embedded human bone marrow cells of acute myeloid leukemia (AML) and myelodysplastic syndrome (MDS) patients with IDH1 mutation. a. AML1, b. MDS-EB1. Scale bars, 5 µm. (**E**) Representative immunofluorescence images of IDH1 (green), GPA (red), and Hoechst 33342 (blue) staining of the 293T cells. Scale bars, 5 µm. (**F**) Representative western blotting images showed the protein expression level of IDH1 on nucleus and cytoplasm of terminal erythroid cells, 293T cells, human HUDEP2 cell lines, K562 and HEL cell lines. RCC1 was used as nuclear loading control, while Tubulin was used as cytoplasm loading control.

The online version of this article includes the following source data and figure supplement(s) for figure 2:

**Source data 1.** Original western blots for *Figure 2F*, indicating the relevant bands.

**Source data 2.** Original files for western blot analysis displayed in *Figure 2F*.

**Figure supplement 1.** Isocitrate dehydrogenase 1 (IDH1) localizes to nucleus during human erythropoiesis.

chromatin structure and function (*Zheng et al., 2020*; *Sun et al., 2022*). Moreover, it has been well established that dynamic changes in methylation and demethylation of histone proteins could modify their interaction with DNA, consequently changing the ratio of heterochromatin and euchromatin (*Grewal and Moazed, 2003*). Thus, we next detected histone modification marks using immunofluorescence imaging and western blotting. For most histone modifications, such as H3K36me2, H3K4me3, H3K4me2, and H3K36me3, there were no significant differences between IDH1-deficient groups and control group on day 15 (*Figure 4—figure supplement 1A, B*). However, abundance and subcellular location of specific histone modifications altered dramatically, including H3K27me2, H3K79me3, and H3K9me3. In control group, the majority of histone modification markers were released to cytoplasm for final degradation. While in IDH1-deficient groups, H3K27me2, H3K79me3, and H3K9me3 were still arrested in nucleus (*Figure 4D–F*). Taken together, all these results strongly suggested that nuclear IDH1 played critical roles in maintaining chromatin state dynamics by affecting accumulation and distribution of specific histone modifications.

## Identification of H3K79me3 as the critical factor in response to IDH1 deficiency

Previous studies have reported that mutant-IDH1 induced chromatin state altering and thus drive development of gliomas and other human malignancies. To further define chromatin state reprogramming induced by IDH1 deficiency during erythropoiesis, we used chromatin immunoprecipitation followed by ChIP-seq to investigate genome-wide distribution of three key histone modifications in IDH1-deficient groups on day 15. First, heatmaps and corresponding profile plots of 3 replicates for ChIP-seq data showed that there was the greatest increase in recruitment of H3K79me3 to chromosome (*Figure 5A and B*). In addition, there were 21,169 peaks of H3K79me3 identified, which were significantly larger than 91 peaks of H3K27me2 and 1740 peaks of H3K9me3 (*Figure 5C*). We further analyzed the peak localization of H3K79me3, H3K27me2, and H3K9me3. Around 45% peaks of H3K79me3 were located in promoter regions. While for H3K27me2 and H3K9me3, peaks covered a large proportion in the distal intergenic region, approximately ~70% (*Figure 5D*). In fact, there were 8602 peaks with H3K79me3 in promoter region, which was ~800 fold more than that of H3K27me2 and H3K9me3 (*Figure 5E*). We subsequently conducted Gene Ontology (GO) analysis on the differential peaks. GO terms from biological processes, cellular components, and molecular functions consistently showed that H3K79me3 peaks were predominantly enriched with promoters of genes involved in RNA splicing and chromatin modification pathway (*Figure 5F*). Taken together, these results indicated that IDH1 deficiency reshaped chromatin states by altering the abundance and distribution of H3K79me3, which was the mechanism underlying roles of IDH1 in modulation of terminal erythropoiesis.

We further detected location of H3K79me3 in control and Sg-NES-IDH1 HUDEP-2 cell line and found that deletion of nuclear IDH1 induced H3K79me3 accumulation in nucleus (*Figure 4—figure supplement 2A*). The production of erythrocytes with abnormal nuclei and the reduction of mature erythrocytes due to IDH1 deletion are prominent features of MDS and AML. In addition, pathogenesis of AML/MDS is significantly associated with IDH1 mutation (*Goasguen et al., 2018*). Therefore, we

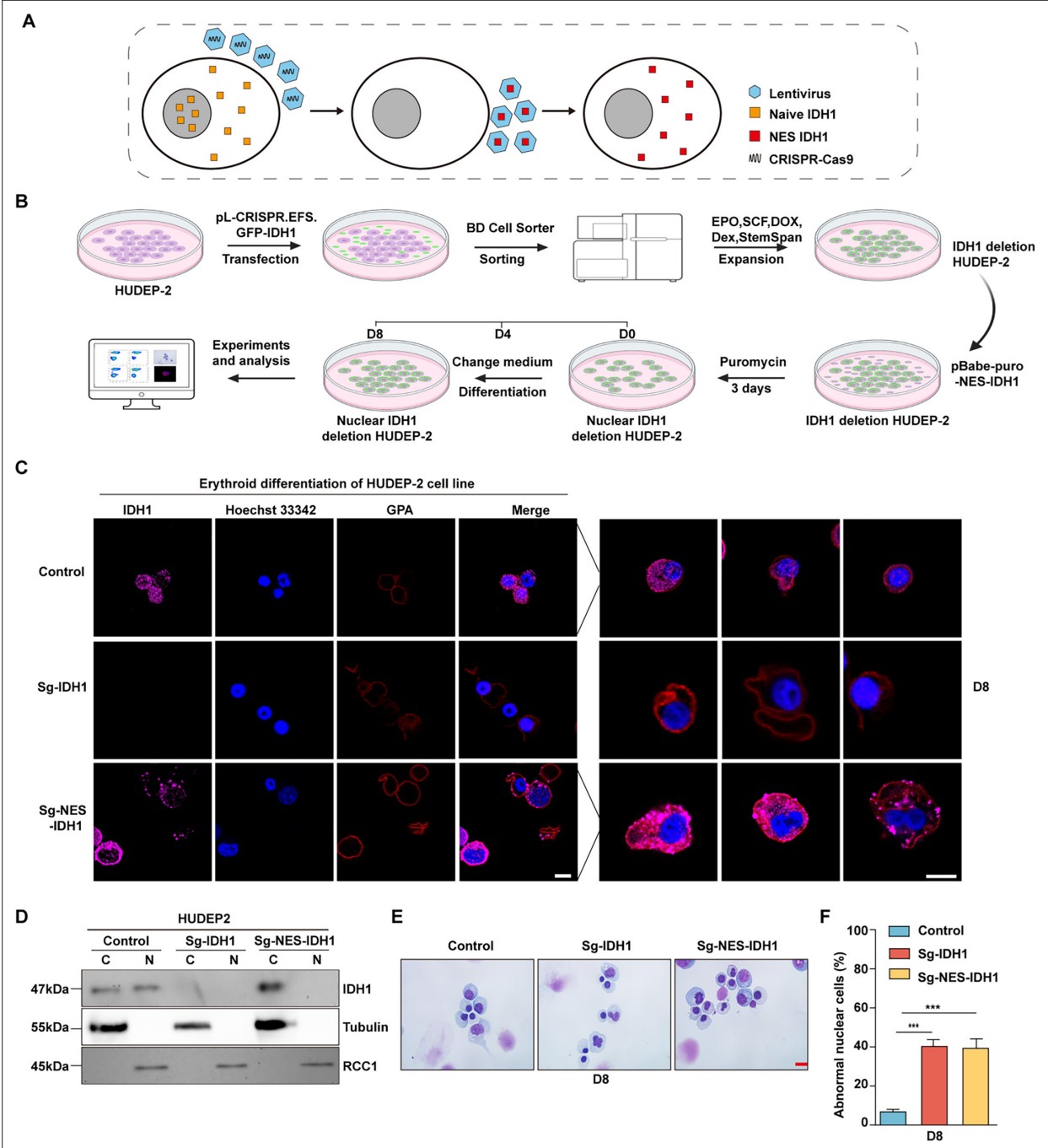

**Figure 3.** Nuclear isocitrate dehydrogenase 1 (IDH1) deletion increased abnormal nuclear cells. (**A**) Schematic diagram of selectively knockdown nuclear IDH1. (**B**) A working model for the construction of Sg-IDH1 HUDEP2 cell line and Sg-NES-IDH1 HUDEP2 cell line. (**C**) Representative immunofluorescence images of location of IDH1 at D8 in HUDEP-2 cell lines. IDH1 (purple), GPA (red), and Hoechst 33342 (blue). Scale bars, 5 μm. (**D**) Representative western blotting images showed the expression level of IDH1 at D8 in the HUDEP-2 cell lines. (**E**) Representative cytospin images of control, Sg-IDH1 HUDEP2 cell line and Sg-NES-IDH1 HUDEP2 cell line. Scale bars, 5 μm. (**F**) Quantitative analysis of the percentage of the abnormal nucleus. Statistical analysis is from three independent biological experiments, and the bar plot represents mean ± SD of triplicate samples. Not significant (ns), *p<0.05, **p<0.01, ***p<0.001.

The online version of this article includes the following source data and figure supplement(s) for figure 3:

**Source data 1.** Original western blots for *Figure 3D*, indicating the relevant bands.

**Source data 2.** Original files for western blot analysis displayed in *Figure 3D*.

**Figure supplement 1.** Knockout nuclear isocitrate dehydrogenase 1 (IDH1) lead to cell number decrease of HUDEP2 cells.

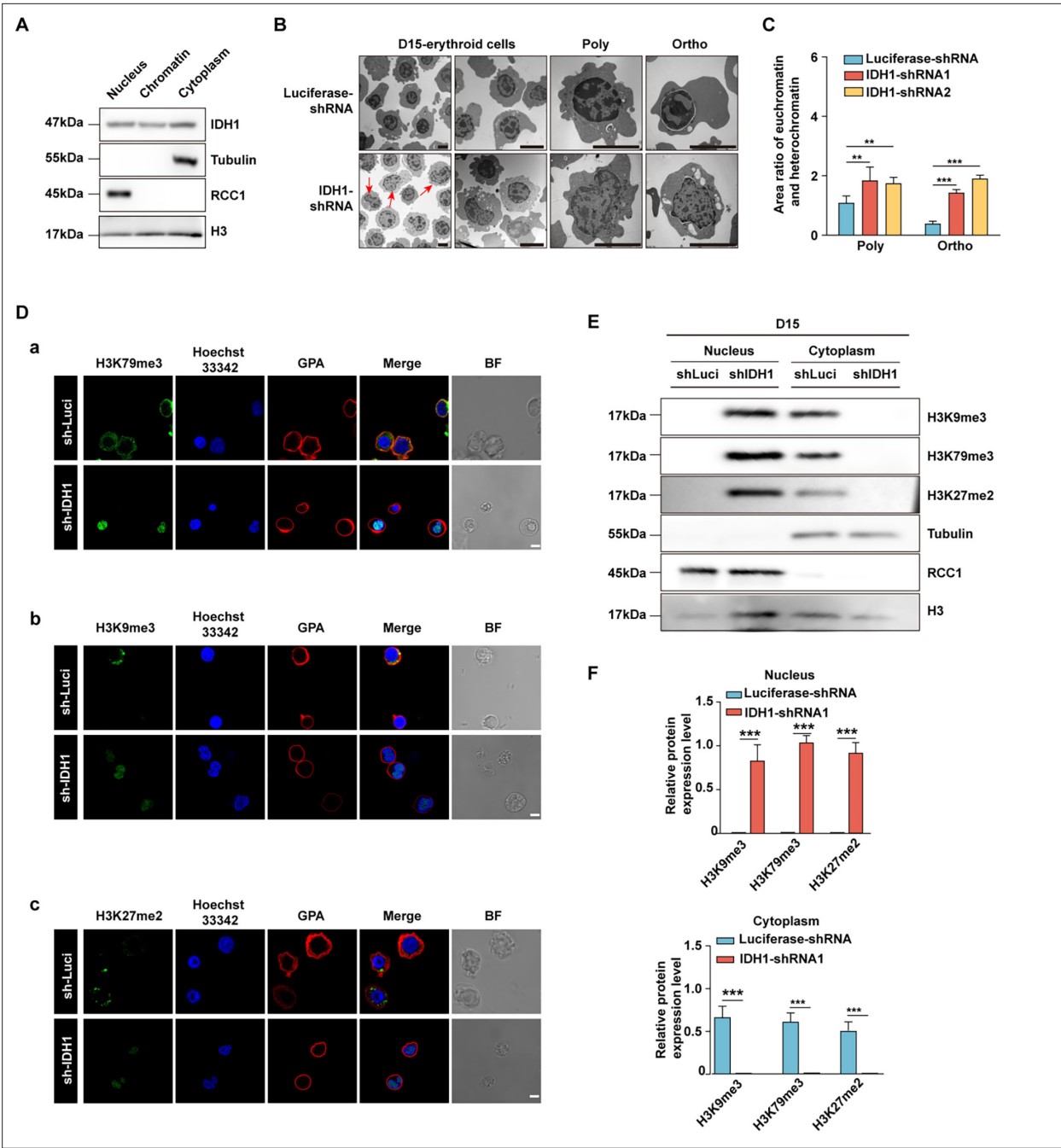

**Figure 4.** Deficiency of isocitrate dehydrogenase 1 (IDH1) reshape chromatin landscape. (**A**) Representative western blotting images showed the expression level of IDH1 in nucleus, chromatin, and cytoplasm of erythroid cells cultured on day 15. (**B**) Representative transmission electron microscopy images showed the distribution of euchromatin and heterochromatin in nuclear on day 15 erythroid cells. (**C**) Quantitative analysis showed the area ratio of euchromatin and heterochromatin from three independent experiments. (**D**) Representative immunofluorescence images of (**a**) H3K79me3 (green), (**b**) H3K9me3 (green), (**c**) H3K27me2 (green). GPA (red) and Hoechst 33342 (blue) staining of Luciferase-shRNA and IDH1-shRNA erythroid cells cultured on day 15. Scale bar, 10 μm. (**E**) Representative western blotting images showed the abundance of H3K79me3, H3K9me3, H3K27me2 in nucleus and cytoplasm of Luciferase-shRNA and IDH1-shRNA erythroid cells cultured on day 15. (**F**) Quantitative analysis of the abundance of H3K79me3, H3K9me3, H3K27me2 in nucleus (above) and cytoplasm (below) of Luciferase-shRNA and IDH1-shRNA erythroid cells cultured on day 15 from three independent biological experiments. The bar plot represents mean ± SD of triplicate samples. Not significant (ns), *p<0.05, **p<0.01, ***p<0.001.

The online version of this article includes the following source data and figure supplement(s) for figure 4:

**Source data 1.** Original western blots for *Figure 4A and E*, indicating the relevant bands.

**Source data 2.** Original files for western blot analysis displayed in *Figure 4A and E*.

*Figure 4 continued on next page*

*Figure 4 continued*

**Figure supplement 1.** Isocitrate dehydrogenase 1 (IDH1) deficiency induced aberrant distribution and accumulation of histone modifications.

**Figure supplement 1—source data 1.** Original western blots for *Figure 4—figure supplement 1A*, indicating the relevant bands.

**Figure supplement 1—source data 2.** Original files for western blot analysis displayed in *Figure 4—figure supplement 1A*.

**Figure supplement 2.** The location of isocitrate dehydrogenase 1 (IDH1) and H3K79me3 in terminal erythroblasts of sg-NES-IDH1 HUDEP-2 cell line and AML/MDA patients.

speculated that aberrant accumulation and distribution of H3K79me3 caused by dysfunction of IDH1 were also another characteristic of MDS/AML bearing IDH1 mutation. Thus, we performed immunofluorescence microscopy observation to further investigate the clinical significance of H3K79me3 in progression of IDH1-mut AML/MDS (*Supplementary file 1*). Notably, an interesting finding was that H3K79me3 colocalized with IDH1 mutants in nucleus (*Figure 4—figure supplement 2B*).

## IDH1 deletion increased chromatin accessibility in late-stage erythroblasts

Histone modifications are important epigenetic markers involved in multiple cellular processes via regulation of gene transcription or remodeling of chromatin structure (*Strahl and Allis, 2000*; *Tan et al., 2011*). Since previous studies have identified H3K79me3 as a transcriptional activating marker (*Shah and Henriksen, 2011*), we speculated that accumulation of H3K79me3 in IDH1-deficient erythroblasts would subsequently lead to switch of closed chromatin into open chromatin. To test our hypothesis, we performed an Assay for Transposase-Accessible Chromatin with high-throughput sequencing (ATAC-Seq) to identify alterations in chromatin accessibility (*Figure 6—figure supplement 1A, B*). Heatmaps and corresponding profile plots displayed chromatin accessibility of Luciferase-shRNA and IDH1-shRNA at peak centers. We found that IDH1-deficient cells showed higher ATAC peak signals compared to control cells (*Figure 6A and B*). Volcano plot showed that there were 2,637 ATAC peaks gained and 442 ATAC peaks lost in presence of IDH1 deficiency (*Figure 6C*). Analysis of localization of ATAC peaks showed that differential peaks were mainly located in promoter regions (*Figure 6D*; *Figure 6—figure supplement 1C*). GO analysis showed that gained ATAC peaks were mainly enriched with promoters of genes involved in various chromatin regulation pathways (*Figure 6E*). To identify which specific DNA binding motifs were enriched in differential peaks, we performed motif analysis using HOMER and found that gained ATAC promoter peaks were enriched with KLF1 binding motifs (p<0.001), which was identified as an erythroid-specific transcription factor (*Figure 6F*). Therefore, these results further confirmed that nuclear IDH1 plays a critical role in chromatin structure modulation as determined by accumulation and distribution of H3K79me3.

## Integrated analysis of ChIP-seq, ATAC-seq, and RNA-seq identified SIRT1 as one of the key genes affected by IDH1 deficiency

To further elucidate mechanisms underlying effects of IDH1 deficiency on changes in chromatin landscape and transcriptional state, we conducted integration analysis of ChIP-seq, ATAC-seq and RNA-seq. RNA-seq analysis was performed to get widespread gene expression in control and IDH1-deficient erythroid cells (*Figure 7—figure supplement 1A*). Volcano plot and heatmap of differentially expressed genes revealed that 3543 genes were upregulated and 3295 genes were downregulated after IDH1 knockdown (*Figure 7—figure supplement 1B, C, and E*). GO analysis showed that upregulated genes mainly enriched in pathway associated with chromatin, such as chromatin organization and chromatin remodeling (*Figure 7—figure supplement 1D*). We further displayed chromatin-related genes, including *TSPYL1*, *SIRT1*, and others (*Figure 7—figure supplement 1F*). Further integrated analysis of RNA-seq and ChIP-seq showed that gene expression levels of H3K79me3-marked genes were upregulated in IDH1-deficient cells (*Figure 7—figure supplement 2A, B*). Furthermore, integrated analysis of ChIP-seq and ATAC-seq confirmed that IDH1 loss resulted in a greater proportion of open chromatin regions in H3K79me3-enriched sites (*Figure 7—figure supplement 2C*). Gene Set Enrichment Analysis (GSEA) showed that chromatin structure modulating pathway, including chromatin silencing and gene expression epigenetics (*Figure 7A*). We found that there were 93 genes overlap in ATAC-seq, ChIP-seq, and RNA-seq (*Figure 7B*), of which three genes shared in chromatin-associated pathways, including *SIRT1*, *NUCKS1*, and *KMT5A* (*Figure 7C*). Transcription factors (TFs)

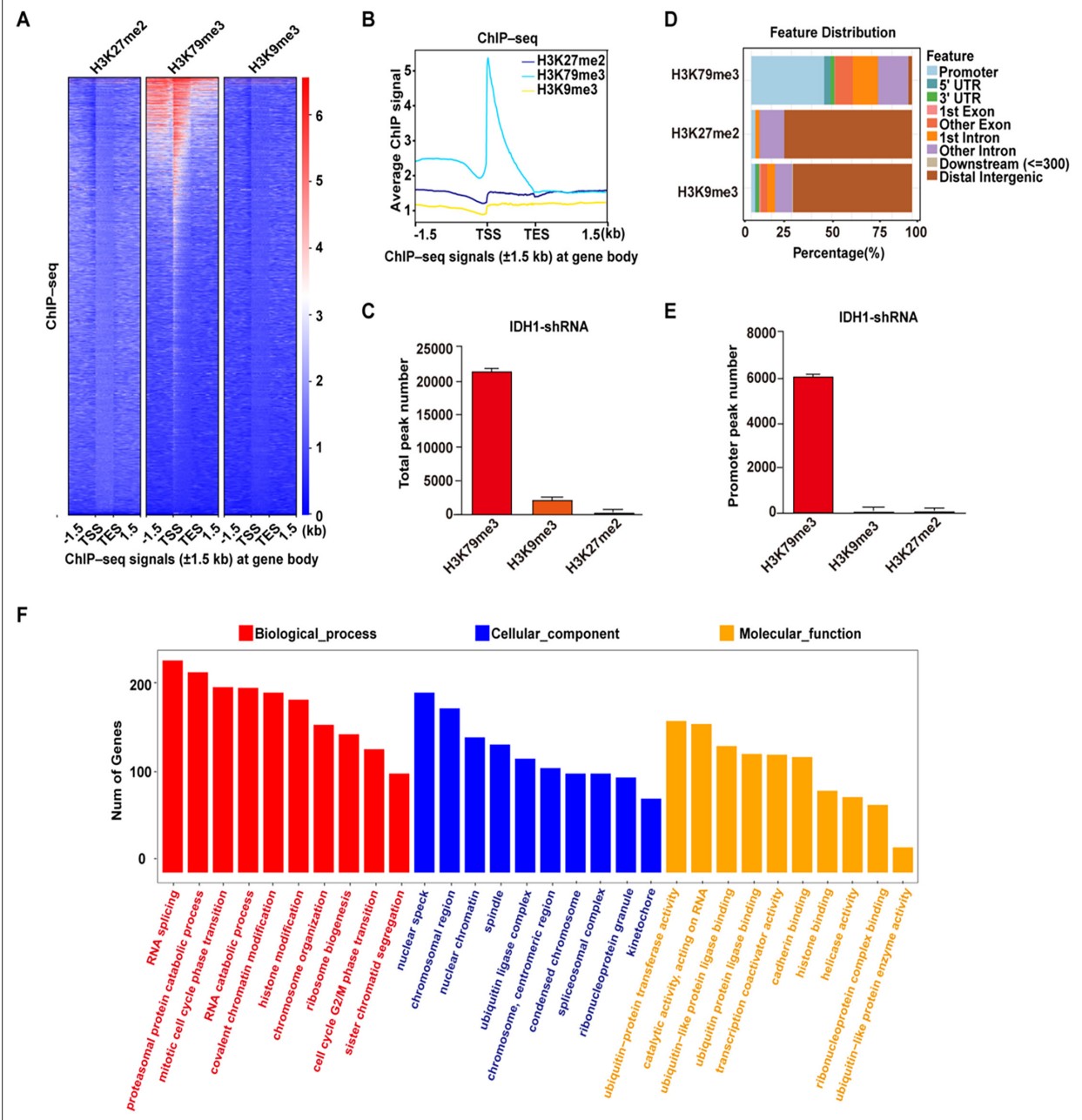

**Figure 5.** Identification of H3K79me3 as the critical factor in response to isocitrate dehydrogenase 1 (IDH1) deficiency. (**A**) Heatmaps displayed normalized ChIP signal of H3K27me2 (left), H3K79me3 (middle), and H3K9me3 (right) at gene body regions. The window represents ±1.5 kb regions from the gene body. TES, transcriptional end site; TSS, transcriptional start site. (**B**) Representative peaks chart image showed normalized ChIP signal of H3K27me2 (cyan), H3K79me3 (blue), and H3K9me3 (yellow) at gene body regions. (**C**) Statistics analysis of total peak number of H3K27me2, H3K79me3, and H3K9me3 from three independent biological experiments. (**D**) The bar plot showed the distribution of ChIPseeker-derived annotations of the genomic loci covered by peaks of H3K79me3, H3K27me2, and H3K9me3. (**E**) Statistics analysis of promoter peak number of H3K27me2, H3K79me3, and H3K9me3 from three independent biological experiments. (**F**) The bar plot showed GO enrichment analysis of the H3K79me3 peaks linked gene promoter.

play key roles in regulation of transcription by recognizing and binding to target gene promoter region. Due to deficiency of IDH1, gained ATAC promoter peaks were enriched with KLF1 binding motifs (*Figure 6F*). IDH1 deficiency caused alteration of chromatin landscape, characterized by aberrant accumulation and distribution of H3K79me3, which led to dysregulation of terminal erythropoiesis. Thus, we speculated that H3K79me3 was involved in SIRT1 up-regulation. Representatively,

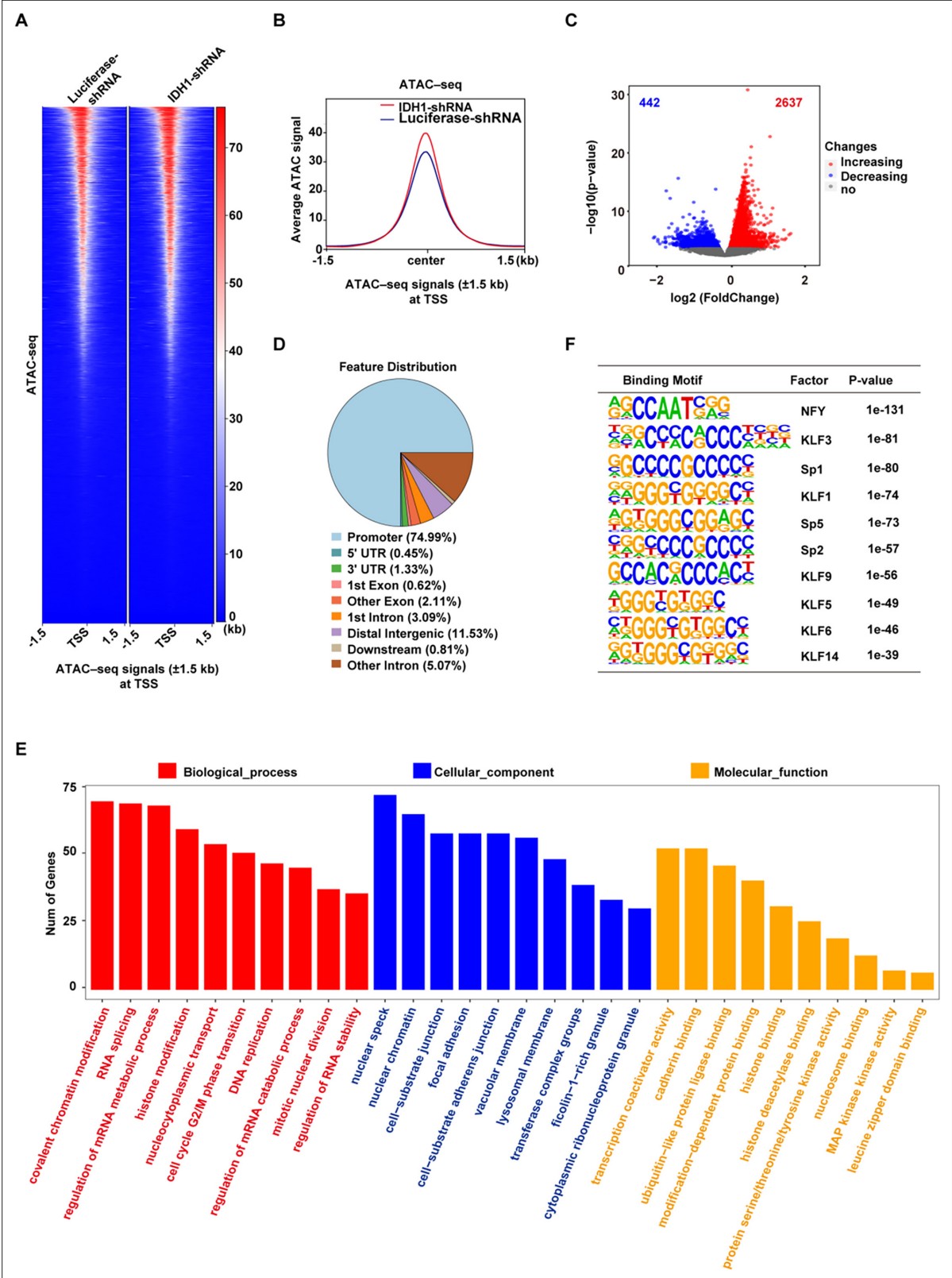

**Figure 6.** Isocitrate dehydrogenase 1 (IDH1) deletion increase the chromatin accessibility in late-stage erythroid cells. (**A**) Heatmaps displayed assay for transposase-accessible chromatin (ATAC) signal of Luciferase-shRNA (left) and IDH1-shRNA (right) at transcriptional start site (TSS). The window represents ±1.5 kb regions from the TSS. (**B**) Representative peaks chart image showed ATAC signal of IDH1-shRNA (green) and Luciferase-shRNA (blue) at TSS. (**C**) The volcano map showed differentially accessible peaks of gain (red color) and loss (blue color). (**D**) The bar plot displayed the distribution

*Figure 6 continued on next page*

*Figure 6 continued*

of peaks relative to gene features for differentially accessible peaks. (**E**) The bar plot showed GO enrichment analysis of the gained peaks linked gene promoter. (**F**) The top regulatory protein-binding sites identified by the HOMER algorithm from differentially accessible peaks. The top 10 motifs were ranked by p-value.

The online version of this article includes the following figure supplement(s) for figure 6:

**Figure supplement 1.** assay for transposase-accessible chromatin with high-throughput sequencing (ATAC-seq) analysis.

gene expression levels and ATAC peak signals at SIRT1, *NUCKS1, and KMT5A*locus were elevated in IDH1-shRNA group and were accompanied by enrichment of H3K79me3 (*Figure 7D*). KLF1 has been characterized as one of the most significant TFs involved in regulation of human erythropoiesis. We thus detected TFs bound on *SIRT1, NUCKS1, and KMT5A* locus. ChIP-seq results showed that KLF1 could bind to promoter regions *SIRT1, NUCKS1 and KMT5A* with the highest peak signal at promoter of *SIRT1* (*Figure 7E*). Using DNA pull-down assay, we further confirmed that H3K79me3 binds to *SIRT1* gene promoter region in response to IDH1 knockdown (*Figure 7F*). These findings strongly suggested that IDH1 deletion can lead to increased accumulation of H3K79me3 in *SIRT1* promoter region and thus switch region into open state, thereby recruiting KLF1 to promote *SIRT1* expression.

## SIRT1 plays a critical role in mediating the regulatory effect of IDH1 during terminal stage erythropoiesis

Further integrated analysis also provided evidence to support our forementioned findings. As shown in *Figure 7F*, RNA-seq, ATAC-seq, and ChIP-seq signal tracks annotated to SIRT1 gene loci showed H3K79me3/ATAC-seq overlap and corresponding upregulation of SIRT1 in IDH1 knockdown group. Based on these results, we speculated that SIRT1 was the key factor that mediated the role of IDH1 in regulation of terminal erythropoiesis. Therefore, we performed verification and rescue experiments by treatment of normal erythroid cells with a SIRT1 activator or treatment of IDH1-deficient cells with a SIRT1 inhibitor, respectively. First, we confirmed that the application of SRT1720 on normal erythroid cells and treatment of IDH1-deficient erythroid cells with SIRT1 inhibitor have no effect on terminal differentiation and did not induce apoptosis (*Figure 8—figure supplements 1 and 2*). The addition of SRT1720 led to nuclear malformations in a dose-dependent manner, in approximately 40% of cells (*Figure 8A*). Importantly, we observed a significant enucleation reduction after addition of SRT1720. In control group, the percentage of enucleated cells was 30% and 45% on day 13 and day 15, respectively, compared to <20% in SRT1720 group (*Figure 8B*). Therefore, SIRT1 activation mimicked the cell dysfunction caused by IDH1 knockdown. Morphological images of IDH1-deficient erythroid cells treated with EX527 showed that malformed nuclei were partially rescued (*Figure 8C*). In addition, treatment with SIRT1 inhibitor EX527 also partially increased enucleation efficiency to 25% and 35% on day 13 and day 15, respectively (*Figure 8D*). In conclusion, these findings prove that SIRT1 is a key regulatory factor that mediates roles of IDH1 in regulation of nuclear morphology and enucleation of terminal stage erythropoiesis.

## Discussion

The effective diagnosis and treatment of tumors bearing IDH1 mutations relies on an understanding of the fundamental biological and physiological roles of IDH1. In this study, we delineated an enzyme activity-independent role of IDH1 in regulation of human erythropoiesis through remodeling chromatin state. We demonstrated that deficient nuclear IDH1 led to dramatic accumulation of multi-histone modifications, among which H3K79me3 was identified as the crucial factor, resulting in SIRT1 upregulation and consequently leading to defects in various critical cell events during terminal stage erythropoiesis.

Recent advances in understanding non-catalytic activity of metabolism enzymes have provided insights into roles of IDH1 (*Thomas et al., 2022*). Change in subcellular distribution of metabolism enzymes is associated with distinct non-canonical functions. It has been reported that metabolic intermediates such as acetyl CoA, α-KG, S-adenosylmethionine, and nicotinamide adenine dinucleotide can act as cofactors of epigenetic modification of nuclear genes to regulate stem cell function and differentiation (*Gut and Verdin, 2013*). Interestingly, as a metabolic enzyme, hexokinase 2 (HK2)

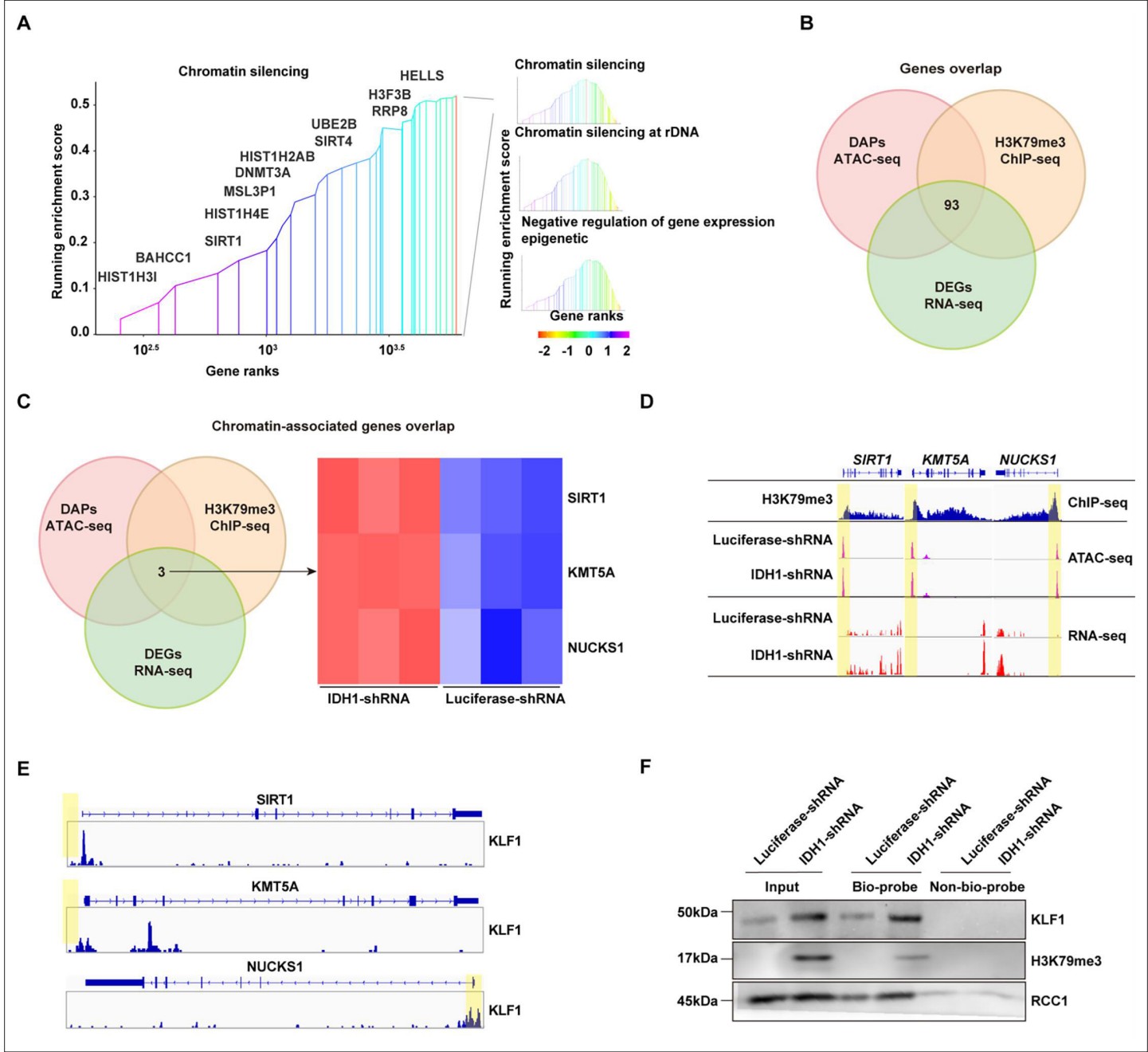

**Figure 7.** Integrated analysis of ChIP-seq, assay for transposase-accessible chromatin with high-throughput sequencing (ATAC-seq), and RNA-seq. (**A**) Gene set enrichment analysis (GSEA) howed chromatin-associated pathways from differentially expressed genes (DEGs) with promoter region marked by H3K79me3. (**B**) Gene overlap analysis of ATAC-seq, ChIP-seq, and RNA-seq. (**C**) Chromatin-associated genes overlap analysis of ATAC-seq, ChIP-seq, and RNA-seq. (**D**) *SIRT1, KMT5A, and NUCKS1* gene locus. Patterns of H3K79me3 modification denoted by ChIP peaks (red) are apparent in IDH1-shRNA increased chromatin accessibility (identified by ATAC-seq) (orange) and gene expression (identified by RNA-seq) (blue). (**E**) KLF1 binding sites of *SIRT1, KMT5A, and NUCKS1* locus. (**F**) DNA pull-down assay showed KLF1 and H3K79me3 could binding to *SIRT1* gene promoter.

The online version of this article includes the following source data and figure supplement(s) for figure 7:

**Source data 1.** Original western blots for *Figure 7F*, indicating the relevant bands.

**Source data 2.** Original files for western blot analysis displayed in *Figure 7F*.

**Figure supplement 1.** RNA-seq analysis.

**Figure supplement 2.** Integrated analysis of ChIP-seq, assay for transposase-accessible chromatin with high-throughput sequencing (ATAC-seq), and RNA-seq.

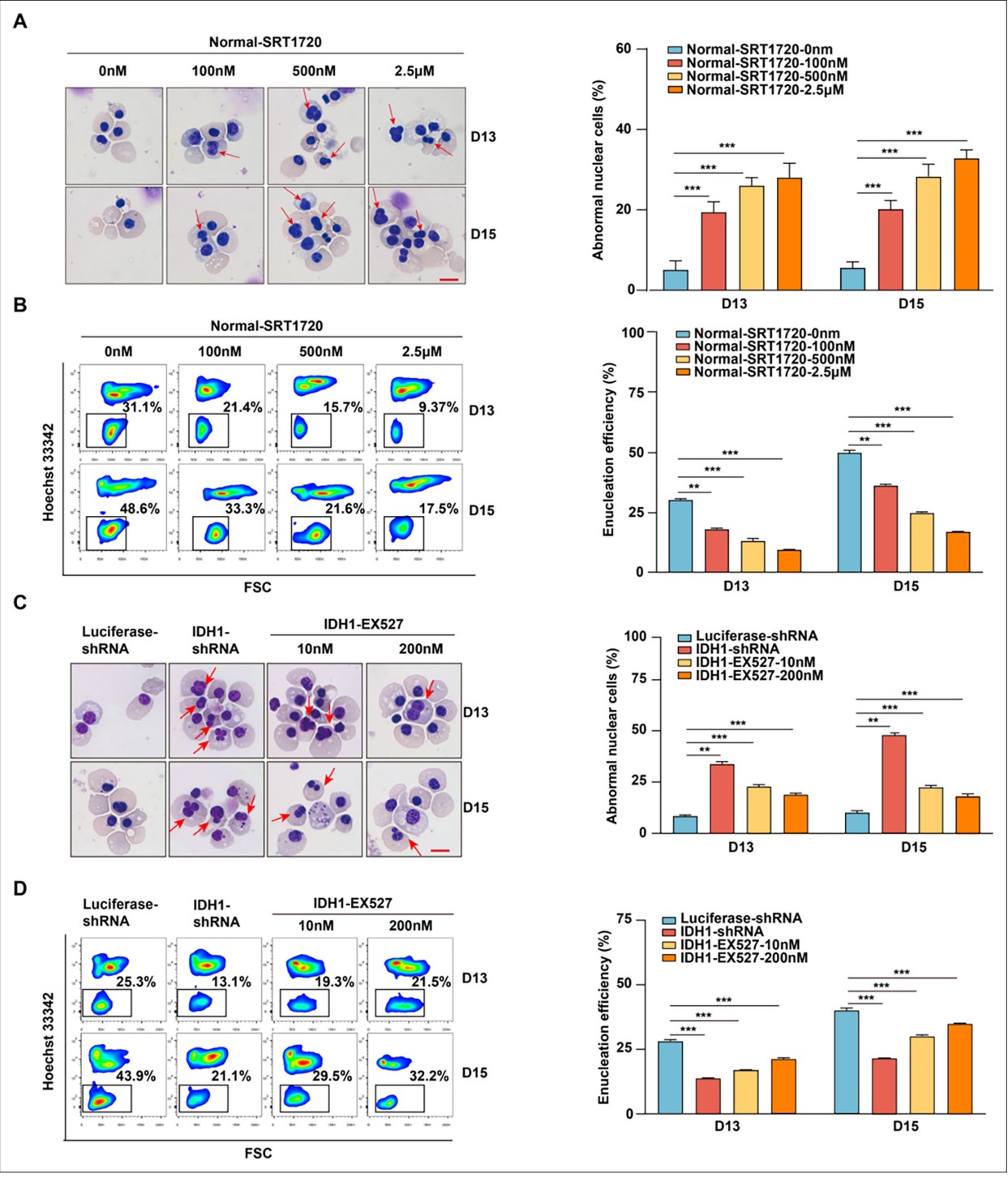

**Figure 8.** SIRT1 plays a critical role in mediating the regulatory effect of isocitrate dehydrogenase 1 (IDH1) during terminal stage erythropoiesis. (**A**) Representative cytospin images of Normal-SRT1720 (0 nM, 100 nM, 500 nM, 2.5 µM) on day 13 and day 15. The red arrows point to the cells with abnormal nucleus. Scale bar, 10 µm. Statistics analysis of abnormal nuclear cells from three independent biological experiments. (**B**) Flow cytometric showing the enucleation efficiency of Normal-SRT1720 (0 nM, 100 nM, 500 nM, 2.5 µM) on day 13 and day 15. Statistics analysis of enucleation efficiency from three independent biological experiments. (**C**) Representative cytospin images of Luciferase-shRNA, IDH1-shRNA, IDH1-shRNA-EX527 (10 nM, 200 nM) on day 13 and day 15. The red arrows point to the cells that are abnormal nucleus. Scale bar, 10 µm. Statistics analysis of abnormal nuclear cells from three independent biological experiments. (**D**) Flow cytometric showing the enucleation efficiency of Luciferase-shRNA, IDH1-shRNA, IDH1-shRNA-EX527 (10 nM, 200 nM) on day 13 and day 15. Statistics analysis of enucleation efficiency from three independent experiments. Statistical analysis

*Figure 8 continued on next page*

*Figure 8 continued*

is from three independent biological experiments, and the bar plot represents mean ± SD of triplicate samples. Not significant (ns), *p<0.05, **p<0.01, ***p<0.001.

The online version of this article includes the following figure supplement(s) for figure 8:

**Figure supplement 1.** Treatment with SIRT1 activator have no effet on cell differentiation and apoptosis of terminal erythroblasts.

**Figure supplement 2.** Treatment with SIRT inhibitor have no effect on cell differentiation and apoptosis of terminal erythroblasts.

localizes to nucleus of AML and normal hematopoietic stem and progenitor cells, and nuclear HK2 has a non-canonical mechanism by which mitochondrial enzymes influence stem cell function independently of their metabolic function (*Thomas et al., 2022*). In addition, glycolytic enzymes that are translocated to nucleus have independent catalytic roles, such as kinase activity and DNA-binding ability that are associated with regulation of gene expression and DNA repair. For instance, nuclear translocated glyceraldehyde 3-phosphate dehydrogenase can bind to E3 ubiquitin ligase SIAH1 and initiate apoptosis, as well as bind to telomeres to maintain their length (*Yego and Mohr, 2010*). In this study, we found that IDH1, as one of key metabolism enzymes, localizes to nucleus in erythroid cells of healthy person and AML/MDS patients, and plays critical roles in maintaining cell morphology and chromatin dynamics in non-canonical functions manner.

Although a few studies have reported that IDH1 is located in nucleus, most studies have focused on the metabolism and RNA binding of IDH1 located in cytoplasm (*Liu et al., 2019*). In line with previous findings, our study confirmed the critical role of IDH1 in regulation of human erythropoiesis, which could not be rescued by addition of ROS scavengers or supplementation with α-KG (*Gonzalez-Menendez et al., 2021*; *Gu et al., 2021*). Previous work reported that IDH1 regulate human erythropoiesis modulating metabolism, based on the finding that defective erythropoiesis induced by IDH1 deficiency could be reversed by application of vitamin C. It should be noted that Vitamin C is a critical metabolic regulator that plays important biological and physiological roles, as well as epigenetic modulatory roles, but underlying mechanism still remains to be well illustrated (*Zhang et al., 2019*; *Mingay et al., 2018*). Based on diverse outcomes of vitamin C treatment and metabolism restoration, we proposed that IDH1 might elicit its effect partially in an non-canonical functions.

Interestingly, we found that IDH1 deficiency had no obvious effect on differentiation and proliferation of proerythroblasts and basophils but affected unique cell events of polychromatic and orthochromatic erythroblasts, including cell morphology, chromatin condensation, and enucleation. Terminal stage erythropoiesis is a highly dynamic and complex process during which chromatin structure undergoes significant changes (*Palis, 2008*). Current and previous studies showed that in addition to chromatin condensation, majority of histones were released into cytoplasm for decomposition during late stage of terminal erythropoiesis and is crucial for chromatin condensation and subsequent enucleation (*Hattangadi et al., 2014*). Chromatin reorganization, histone release, and redistribution are essential for chromatin reprogramming on terminal erythroid development. In present study, our findings showed that IDH1 deficiency led to accumulation of multiple histone modification markers in nucleus, particularly in polychromatic and orthochromatic cells, might explain effects of IDH1 deficiency at specific development stages. In addition, we found that IDH1 deficiency caused methylated histones accumulation on chromatin and increased accessibility to generate open chromatin state for transcription activation, which further exacerbates defective terminal erythropoiesis. Present study is first to report that nuclear IDH1 has important roles in maintenance of dynamic changes in chromatin states. Our findings suggest synergistic actions of nuclear IDH1 and cytoplasmic on regulation of human erythropoiesis by modulating metabolism and chromatin architecture.

Our results also suggested that IDH1 deficiency was related to aberrant accumulation of H3K79me3 and SIRT1 upregulation. Although disturbed methylation catalyzed by DOT1 is a well-recognized target for diagnosis and treatment of leukemia and its transformation, the majority of previous studies mainly focused on the role of H3K79 demethylation (*Godfrey et al., 2021*; *Godfrey et al., 2019*; *Li et al., 2018*). In this study, in vitro and in vivo (patient samples) experiments showed that IDH1 deficiency led to significant H3K79me3 accumulation, which could be used to diagnose dyserythropoiesis with IDH1 functional defect. SIRT1 is important for catalyzing removal of acetyl groups from non-histone and histone proteins. Previous studies have demonstrated that SIRT1 is involved in a wide range of physiological functions, including regulation of gene expression, cell growth, DNA

repair (*Kuno et al., 2022*), oxidative stress (*Liu and Shi, 2022*), metabolism (*Zhang et al., 2023*), and aging (*Rajendran et al., 2011*). In addition, SIRT1 activates expression of fetal hemoglobin genes in adult human erythroblasts (*Dai et al., 2017*). Notably, SIRT1 is also involved in regulation of higher-order chromatin structure and various pathways related to pathogenesis and transformation of various dyserythropoiesis-related hematopoietic diseases (*Li et al., 2014*). Overexpression and underexpression of SIRT1 are strongly associated with pathogenesis of AML and MDS. Both inhibitory and activatory strategies have been proposed for diagnosis and treatment of these diseases (*Huang et al., 2019*). In current study, we proved that IDH1 deficiency leads to significantly increased SIRT1 expression. Series of phenotypes caused by IDH1 deficiency, such as nuclear malformation, aberrant nuclear condensation, and impaired enucleation, could be partially rescued by addition of specific SIRT1 inhibitor. These findings confirm that inhibitory strategy is promising for diagnosis and treatment of dyserythropoiesis-related diseases, particularly in patients with IDH1 mutation.

In conclusion, our work provides a novel insight into the role of IDH1 in the regulation of chromatin states. The findings related to IDH1-H3K79me3-SIRT1 regulatory axis indicates that H3K79me3 and SIRT1 may be targeted for the diagnosis and treatment of diseases with IDH1 mutations.

## Materials and methods

### Human samples

Human bone marrow samples were obtained from 10 AML/MDS patients with IDH1 mutation (4 MDS and 6 AML) in the Department of Hematology, First Affiliated Hospital of Zhengzhou University (Zhengzhou, China). Details of the samples are provided in *Supplementary file 1*. We enrolled previously untreated AML and MDS patients with IDH1 R132 mutations in a multicenter, double-blind, randomized controlled trial. All cases met WHO 2022 diagnostic criteria with IDH1 mutation status confirmed by digital PCR (*Pollyea et al., 2021*; *Stahl et al., 2023*). Written informed consent was obtained from all participants, and the protocols were approved by the Ethics Committee of The First Affiliated Hospital of Zhengzhou University (2021-KY-0575–002).

### Generation of cell lines

The K562 and HEL cell lines were obtained from The State Key Laboratory of Medical Genetics & School of Life Sciences, Central South University (Changsha, China). Human HUDEP2 cell line was obtained from the Institute of Hematology & Blood Diseases Hospital, Chinese Academy of Medical Sciences & Peking Union Medical College (Tianjin, China). The detailed methods have been described previously including culture medium composition, the culture protocol (*Trakarnsanga et al., 2017*).

Nuclear IDH1 was deleted in the HUDEP2 cell line using the following approach: sgRNA sequences targeting different genomic regions of IDH1 (IDH1-sgRNA-Top: CACCGATGTAGATCCAATTCCACG T and IDH1-sgRNA-Bottom: AAACACGTGGAATTGGATCTACATC) were annealed and ligated with pL-CRISPR.EFS.GFP vector by restriction digest with BsmBI (*Heckl et al., 2014*). Production of lentiviral vectors was performed according to standard protocol. HUDEP2 cells were then infected with viral particles carrying sgRNA sequences targeting IDH1 and sorted with GFP to get Sg-IDH1 HUDEP2 cells.

pBabe-puro-NES-IDH1 was generated using primers encoding for the PKI NES (LALKLAGLDI) (*Wen et al., 1995*). Forward (TCTAGGCGCCGGCCGGATCCCGCCACCATGTCC-AAAAAAATCAGT GGCGG) and reverse (ACCACTGTGCTGGCGAATTCTTAAA-TATCCAGGCCCGCCAGTTTCAGCG CCAGGTCGACAAGTTTGGCCTGAGCTAGTTT) primers were annealed and ligated with pBabe-puro by restriction digest with BamHI and EcoRI (*Morgenstern and Land, 1990*). Production of lentiviral vectors was performed according to standard protocol. Sg-IDH1 HUDEP2 cells were infected with viral particles of pBabe-puro-NES-IDH1 and selected with Puromycin to get Sg-NES-IDH1 HUDEP2 cells.

### Culture of CD34⁺ cells and shRNA or siRNA-mediated knockdown of IDH1

Primary human CD34⁺ cells were acquired from cord blood by positive selection using CD34⁺ magnetic selective beads system according to the manufacturer's protocol (*Qu et al., 2018*). The detailed methods have been described previously including culture medium composition, the culture

protocol, preparation of lentivirus (*Li et al., 2023*). The sequence of IDH1-shRNA and IDH1-siRNA in experiments were shown in *Supplementary file 1*.

## Drug treatment

The drugs for cell treatment were as follows. N-Acetyl-L-cysteine (MCE, HY-B0215) used at a final concentration of 10 μM. Glutathione (Selleck Chemicals, S4606) was dissolved in DMSO, and used at a final concentration of 50 μM. α-ketoglutanic acid (MilliporeSigma, K1128) was used at a final concentration of 50 μM. EX527 (Selleck, S1541) used at final concentrations of 10 and 200 nM and SRT1720 (MCE, HY10532) used at final concentrations of 0.1, 0.5 nM, and 2.5 μM.

## Immunofluorescent staining of AML or MDS patients paraffin-embedded bone marrow cells

Formalin-fixed, paraffin-embedded tissues blocks of bone marrow samples from AML and MDS patients bearing IDH1 mutation were sectioned at a thickness of about 4–5 μm. Slides were floated in a 40 °C water bath and then transferred onto glass slides. After drying for 12 hr, slides were deparaffinized with xylene and rehydrated with alcohol. Then, slides were incubated in 10 mM citrate buffer (pH 6.0) at 95–100°C for 10 min. 3% BSA in PBS were added onto slides and incubated at 25 °C for 30 min. Slides were incubated with primary antibodies at 4 °C for 12 hr and with secondary antibody at 25 °C for 30 min. Nuclear was stained with Hoechst 33342 (blue). The following primary antibodies were used as follows: rabbit polyclonal antibody to IDH1(#12332–1-AP, Proteintech), mouse monoclonal antibody to Histone H3 (Tri methyl K79) (BSM-33098M, Thermo Fisher Scientific). Secondary antibodies were goat anti-rabbit or goat anti-mouse labeled with Alexa Fluor 488 or Cy5(Servicebio). Pannoramic MIDI (3DHISTECH) slice scanner was used to acquire images.

## DNA pull-down assay

The sequence of the biotin-labeled probes for SIRT1 was as follows (Forward oligo: TCCCAAAG TGCTGGGATTACAG; Reverse oligo: GCACCTCGGTACCCAATCG). Genomic DNA containing the target DNA sequence was used as a template, the promoter region of SIRT1 was amplified by PCR, including biotin-labeled and non-biotin-labeled. 2 ug probes (target gene biotin-labeled probe set and non-labeled probe set, Empty magnetic beads group) were added in EP tube and incubated for 8 hr in a 4 °C refrigerator. 300 μL cell lysates (empty beads group without lysis solution) were added into magnetic bead-probe mixture and incubated in a 4 °C refrigerator for 12 hr, 30 μL (10%) of the remaining lysates were taken as the input group. Incubation mixture was centrifuged at 3000 rpm for 2 min and washed five times. Loading buffer was added into each group (including input group and beads group). Tubes were input into boiling water bath for 10 min and centrifuged at 3000 rpm for 5 min. The obtained production can be directly used for WB experiment.

## Protein extraction

Total protein extraction: Cells were harvested and washed with PBS. Then, cells were centrifugated at 300×g for 5 min. Cell lysates were prepared with RIPA lysis buffer (#89900, Thermo Fisher Scientific) in the presence of proteinase inhibitor phenylmethanesulfonylfluoride fluoride (PMSF) (#36978, Thermo Fisher Scientific).

Cytoplasm and nuclear protein extraction were used by NE-PER Nuclear and Cytoplasmic Extraction Reagents kit (#78833) from Thermo Scientific. The detailed method was described as follows: Cells (5-10×10$^6$) were harvested and washed three times with PBS. Then fresh cells were centrifugated at 300×g for 5 min. Ice-cold CER I was added to the cells and the tube was vigorously vortexed on the highest setting for 15 s to fully suspend the cell pellet. Then cells were incubated on ice for 10 min. Ice-cold CER II was then added to the tube and vortexed for another 5 s on the highest setting. Then cells were incubated on ice for 2 min. The supernatant was collected after centrifugation for 5 min at 16,000×g and then immediately transferred to a clean pre-chilled tube. Cytoplasmic protein was fully extracted. Then, insoluble fractions were suspended using ice-cold NER and vortexed on the highest setting for 15 s. Samples were placed on ice and oscillated for 15 s every 10 min for 40 min. Finally, the supernatant containing extracted nucleoprotein was immediately transferred to a clean pre-chilled tube. All reagents used during the experiment should be placed on ice. Maintain the volume ratio of CER I: CER II: NER reagents at 200:11:100 μL, respectively. Protein concentration was measured by

BCA protein concentration determination using Micro BCA Protein Assay Kit (23235, Thermo Fisher Scientific).

## Cytospin assay

Obtaining sample slides: cells ($5 \times 10^4$) were collected in a 1.5 mL Eppendorf tube (EP), washed with PBS, and centrifugated at 300×g for 5 min at 4 °C. Cells were resuspended with 200 μL PBS. Then, cells were adhered to slides by cell spin centrifugation. The fresh cytospin were fetched and dried carefully without damage.

Proceeding with May Grunwald/Giemsa staining: Cells were stained with May-Grunwald solution (MG500, Sigma) for 5 min and then washed with 40 mM Tris buffer (pH 7.2) for 90 s. Then, they were dyed with Giemsa solution (GS500, Sigma) for 15 min at once. The images were finally taken using a standard light microscope (Axio Imager.A2, Carl Zeiss Microscopy GmbH, Jena, Germany).

## RNA extraction and quantitative reverse transcription-PCR assays

Total RNA was separated from human erythroblasts at distinct stages using RNA extract kits (#74104, Tiangen Biotech). The concentration was established using the Nanodrop (Thermo Fisher Scientific). RNA samples underwent reverse transcription with HiFi-MMLV cDNA Kit (CW0744M, CWBIO). Quantitative reverse transcription-PCR was completed by using SYBR Select Master Mix (#4472903, Thermo Fisher Scientific) and Light Cycler 480 system (Roche Life Science). Primers were obtained from Harvard primer bank. Relative expression levels were normalized to GAPDH. The primer sequences of *IDH1* and *GAPDH* were shown in **Supplementary file 1**.

## Flow cytometric analysis and fluorescence-activated cell sorting of erythroblasts

The differentiation of erythroid cells was assessed by the expression of surface markers using flow cytometer (*Li et al., 2023*). The erythroid cells at the distinct developmental stage were sorted using a MoFlo Astrios Cell Sorter (Beckman Coulter Life Science) as previously described (*Qu et al., 2018*).

## Immunofluorescence imaging microscopy

Cells ($0.5 \times 10^6$) were collected and washed with 1x PBS. Then, cells were fixed with 1% paraformaldehyde for 15 min and permeabilized with 0.1% Triton X-100 in 0.25% paraformaldehyde-PBS for 10 min. Then, cells were incubated in PBS with 10% horse serum and 0.1% Triton X-100 for 30 min to minimize nonspecific antibody binding. Cells were incubated with primary antibodies at 4 °C overnight, washed three times with PBS, and incubated with the appropriate secondary antibody at room temperature for 30 min. Nuclei were stained with Hoechst 33342 (blue). After washing three times with PBS to remove nonspecific staining. The cells were collected and seeded onto the Thermo Scientific Nunc Lab-Tek II chamber (#155382). Images were collected and visualized under a confocal laser scanning microscope with a 100x oil objective lens (Zeiss LSM780, Carl Zeiss Microscopy GmbH, Jena, Germany).

## Measurement of ROS

DCFH-DA (2,7-Dichlorodihydrofluorescein diacetate) was used to assess reactive oxygen species (ROS) formation using Reactive Oxygen Species Assay Kit (S0033S, Beyotime Biotechnology). Cells ($0.5 \times 10^6$) were collected and washed with PBS buffer. Then, cells were resuspended and incubated in medium containing 10 μM DCFH-DA at 37 °C for 30 min. Cells were washed with PBS three times to remove DCFH-DA that had not entered cells. After entering cells, the DCCFH-DA probe was hydrolyzed by esterase to form DCFH, which was further oxidized by ROS in cells to form fluorescent DCF. The fluorescence intensity was measured using BD LSRFortessa flow cytometry to measure the level of ROS.

## Quantification of α-KG level

α-KG were detected by using an α-KG assay kit (MAK054, Sigma), Preparation of standard substance: α-KG stock solution was diluted with water to 1 mm standard buffer. 0, 2, 4, 6, 8, and 10 μL of the 1 mM α-KG standard buffer were added successively into a 96 well plate, generating 0 (blank), 2, 4, 6, 8, and 10 nmol/well standards. α-KG assay buffer was added to each well to bring the volume to

50 µL. Sample Preparation: Cells ($0.5\times10^6$) were homogenized in 100 µL ice cold α-KG buffer, then were centrifuged at 13,000×g for 10 min to remove insoluble material. Samples were prepared to a final volume of 100 µL reagent, including 94 µL α-KG assay buffer, 2 µL α-KG converting enzyme, 2 µL α-KG development enzyme mix, and 2 µL flourescent peroxidase substrate. Blank control group was similar with the sample preparation group with no α-KG converting enzyme. Then the samples were incubated at 37 °C for 30 min. SpectraMax i3 (Molecular Devices) was used to check the concentration of colorimetric product at 570 nm wavelength to determine the total α-KG concentration.

## Transmission electron microscopy

Cells ($20\times10^6$) were collected and fixed with 2.5% glutaraldehyde. The fixed cells were washed in 0.1 M sodium caccodylate buffer (pH 7.2), and fixed with 1% osmium tetraoxide in sodium caccodylate buffer. Then, cells were dehydrated in ethanol (successively in 70%, 95%, and absolute ethanol), treated with propylene oxide (a transitional solvent), infiltrated in a mixture of propylene oxide and resin (Epon), embedded in pure resin mixture, and cured at 60 °C. Thin sections of 50 nm were generated using an LKB 2088 Ultratome V, applied on copper slot grids, and stained separately with 5% uranyl acetate and 3% lead citrate. Images were generated using a 120-Kv H-7700 Hitachi transmission electron microscope. The area of nuclear, heterochromatin, and euchromatin were quantified using Image J software.

## RNA-seq and bioinformatics analysis

Pre-sequencing experimental procedures: Cells ($2\times10^6$) were collected and washed twice by PBS. RNA was extracted and reverse transcribed into cDNA. The cDNA libraries were prepared using NovaSeq 5000/6000 S4 Reagent Kit (#A00358, Illumina) and sequenced on an Illumina X-ten (E00487, Illumina). The useful Perl script was used to filter the original data (Raw Data) and compare data to the reference genome by using HISAT2 which download from ENSEMBL database (human genome builds GRCh38.87) (*Kim et al., 2015*). Reads Count for each gene in each sample was counted by HTSeq v0.6.0 (*Anders et al., 2015*), and FPKM (Fragments Per Kilobase Million Mapped Reads) was then calculated to estimate the expression level. DESeq2 version 1.6.3 are package for Identifying Differentially Expressed Genes (DEGs) from data (FPKM >10, q<0.05, and fold change >2) (*Anders and Huber, 2010*). Gene Ontology was used to make function enrichment analysis with $q<0.05$. Gene Set Enrichment Analysis (GSEA) was performed using GSEA version 4.1.0 software (*Subramanian et al., 2005*). RNA-seq data have been deposited in the GEO under accession code GSE223141.

## Assay for transposase-Aaccessible chromatin with high throughput sequencing

Pre-sequencing experimental procedures: Cells ($2\times10^6$) were collected and washed twice by PBS. Cells were prepared for detection, DNA transposition, PCR amplification, fragment selection, library quality control, and computer sequencing. Sequencing and analysis: Filter data to get high-quality existing data. After the quality control test is qualified, clean data will be mapped the reference genome which download from ENSEMBL database (human genome builds GRCh38.87) using Bowtie2 (*Langmead et al., 2019*; *Langmead and Salzberg, 2012*). Identifying statistically significant differentially accessible regions using DEseq2 parameters of DiffBind version 3.8.4 (*Ross-Innes et al., 2012*). The ATAC peak/region <1000 bp from the nearest TSS was designated as the promoter. Reads coverage and depth were calculated by samtools version 1.6 software (*Bonfield and Marshall, 2017*; *Li et al., 2009*). Signal track files in Big Wig format were generated using the Deeptools version 2.5.7 and were normalized to 1 million reads for visualization (*Ramírez et al., 2014*). MACS2 version 2.1.1 was used to identify peaks using parameters 'nomodel shift –100 extsize 200' with a q value of <0.05 (*Zhang et al., 2008*). Annotation of peaks were performed using ChIPseeker version 1.20.0 software (*Yu et al., 2015*). GO enrichment analysis and visualization of differentially accessible regions was performed using the clusterProfiler version 4.6.0 with q<0.05 (*Yu et al., 2012*). Overrepresented motif analysis was performed by Hypergeometric Optimization of Motif Enrichment (HOMER) tool Homer version 4.11 (*Heinz et al., 2010*). ATAC-seq have been deposited in the GEO under accession code GSE222401.

## Chromatin immunoprecipitation and sequencing

Sample preparation procedures: cells ($20\times10^6$) were collected and washed twice by PBS, then cells were resuspended in 1% formaldehyde, cross-linked, and quenched with 125 mm glycine in a shaker

at 25 °C for 10 min. Cells were centrifugated 1500×g at 25 °C for 5 min. The precipitates were suspended using pre-cooled PBS and centrifuged at 1500×g for 5 min. Supernatants which around the sediment were removed very carefully.

Pre-sequencing experimental procedures: Firstly, nanodrop, qubit, and Agilent 2100 were used to assess the sample quality, including purity, concentration, and fragment distribution. For the library building process, DNA fragments were repaired at the end, added with basic group A and sequencing connector; then PCR amplification and fragment screening were used to complete the preparation of the library. The qualified library was ready for sequencing.

Sequencing and analysis: ChIP-seq reads were aligned to the reference genome using Bowtie2 version 2.3.5.1, and only uniquely and non-duplicate mapped reads were utilized to perform the downstream analysis. MACS2 version 2.1.1 was used to identify peaks using parameters '–broad–broad-cutoff 0.01' with a q value of <0.05. Annotation of peaks were performed using ChIPseeker version 1.20.0 software. KLF1 raw ChIP-seq data were downloaded from the Gene Expression Omnibus (GEO) GSE104574 (*Choudhuri et al., 2020*). Other analysis methods as described for ATAC-seq. ChIP-seq data have been deposited in the GEO under accession code GSE222296.

## Acknowledgements

This work was supported by grants from the Natural Science Foundation of China (82170116, 81870094, 81570099, and 82300134).

## Additional information

### Funding

| Funder | Grant reference number | Author |
|---|---|---|
| National Natural Science Foundation of China | 82170116 | Lixiang Chen |
| National Natural Science Foundation of China | 81870094 | Lixiang Chen |
| National Natural Science Foundation of China | 81570099 | Lixiang Chen |
| National Natural Science Foundation of China | 82300134 | Fumin Xue |

The funders had no role in study design, data collection and interpretation, or the decision to submit the work for publication.

### Author contributions

Mengjia Li, Hengchao Zhang, Xiuyun Wu, Mengqi Yu, Qianqian Yang, Lei Sun, Wei Li, Zhongxing Jiang, Fumin Xue, Investigation; Ting Wang, Supervision; Xiuli An, Supervision, Writing – review and editing; Lixiang Chen, Supervision, Writing – original draft, Writing – review and editing

### Author ORCIDs

Mengjia Li ![ORCID] https://orcid.org/0009-0002-1170-4946
Hengchao Zhang ![ORCID] https://orcid.org/0000-0001-9873-9986
Xiuyun Wu ![ORCID] https://orcid.org/0000-0003-3607-3191
Mengqi Yu ![ORCID] https://orcid.org/0009-0007-5949-6978
Qianqian Yang ![ORCID] https://orcid.org/0009-0008-0195-833X
Lei Sun ![ORCID] https://orcid.org/0009-0000-8697-7514
Wei Li ![ORCID] https://orcid.org/0000-0001-5550-508X
Zhongxing Jiang ![ORCID] https://orcid.org/0000-0002-0277-4574
Fumin Xue ![ORCID] https://orcid.org/0000-0002-6199-0592
Ting Wang ![ORCID] https://orcid.org/0009-0002-1203-303X
Xiuli An ![ORCID] https://orcid.org/0000-0002-3582-9404
Lixiang Chen ![ORCID] https://orcid.org/0000-0001-7785-2496

### Ethics

This study involving human participants followed the principles outlined in the Declaration of Helsinki and was approved by the Ethics Committee of the First Affiliated Hospital of Zhengzhou University (Ethics Approval No. 2021-KY-0575-002). In addition, all participants gave written consent prior to participation.

Reviewer #1 (Public review): https://doi.org/10.7554/eLife.100406.3.sa1
Reviewer #2 (Public review): https://doi.org/10.7554/eLife.100406.3.sa2
Reviewer #3 (Public review): https://doi.org/10.7554/eLife.100406.3.sa3
Author response https://doi.org/10.7554/eLife.100406.3.sa4

## Additional files

### Supplementary files

Supplementary file 1. Information for human research participants and primer sequence.
MDAR checklist

### Data availability

The data underlying this article are available in article and in its supplementary files. RNA-seq data have been deposited in the GEO under accession code GSE223141. ChIP-seq data have been deposited in the GEO under accession code GSE222296. ATAC-seq have been deposited in the GEO under accession code GSE222401.

The following datasets were generated:

| Author(s) | Year | Dataset title | Dataset URL | Database and Identifier |
|---|---|---|---|---|
| Li M, Chen L | 2025 | Nuclear IDH1 regulates human erythropoiesis by eliciting chromatin state reprogramming [RNA-seq] | https://www.ncbi.nlm.nih.gov/geo/query/acc.cgi?acc=GSE223141 | NCBI Gene Expression Omnibus, GSE223141 |
| Li M, Chen L | 2025 | Nuclear IDH1 regulates human erythropoiesis by eliciting chromatin state reprogramming | https://www.ncbi.nlm.nih.gov/geo/query/acc.cgi?acc=GSE222296 | NCBI Gene Expression Omnibus, GSE222296 |
| Li M, Chen L | 2025 | Nuclear IDH1 regulates human erythropoiesis by eliciting chromatin state reprogramming | https://www.ncbi.nlm.nih.gov/geo/query/acc.cgi?acc=GSE222401 | NCBI Gene Expression Omnibus, GSE222401 |

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
