## [Editor Report · eLife Assessment]

This study by Li et al. presents **important** findings on the metabolism-independent role of nuclear IDH1 in chromatin regulation during erythropoiesis. The authors provide **convincing** evidence that IDH1 deficiency disrupts H3K79 methylation and nuclear architecture, contributing to dyserythropoiesis. Their findings offer invaluable mechanistic insights with potential therapeutic implications for erythroid disorders and hematologic malignancies.

---

## [Referee Report · Reviewer #1 (Public review)]

The manuscript by Li et al., investigates metabolism independent role of nuclear IDH1 in chromatin state reprogramming during erythropoiesis. The authors describe accumulation and redistribution of histone H3K79me3, and downregulation of SIRT1, as a cause for dyserythropoiesis observed due to IDH1 deficiency. The authors studied the consequences of IDH1 knockdown, and targeted knockout of nuclear IDH1, in normal human erythroid cells derived from hematopoietic stem and progenitor cells and HUDEP2 cells respectively. They further correlate some of the observations such as nuclear localization of IDH1 and aberrant localization of histone modifications in MDS and AML patient samples harboring IDH1 mutations. These observations are overall intriguing from a mechanistic perspective and hold therapeutic significance. The authors have addressed the previous concerns sufficiently in the revised manuscript.

---

## [Referee Report · Reviewer #2 (Public review)]

Li, Zhang, Wu, and colleagues investigated the non-canonical localization of IDH1 in the cell nucleus and its unconventional functions, expanding our understanding of the roles of metabolic enzymes such as IDH1. To study its nuclear function, they generated a HUDEP2 cell line with a specific deletion of nuclear IDH1. They found that the loss of nuclear IDH1 led to abnormalities in nuclear morphology and chromatin organization, particularly in H3K79me3. By integrating ChIP-seq, ATAC-seq, and RNA-seq analyses, they identified SIRT1 as a key regulatory factor mediating IDH1's role in nuclear morphology regulation during the terminal stages of erythroid differentiation.

Notably, abnormalities in H3K79me3 were also observed in AML/MDS patients harboring IDH1 mutations, offering new perspectives for disease diagnosis and treatment. To robustly determine the nuclear distribution of IDH1 in erythroid cells, the authors employed multiple approaches, including immunofluorescence and nucleus-cytoplasm fractionation. The development of a HUDEP2 cell line lacking nuclear IDH1 was pivotal for studying its non-canonical nuclear functions.

Experimental results, including euchromatin/heterochromatin observations, histone modification analyses, ChIP-seq, and ATAC-seq, indicated that the deletion of IDH1 disrupts the chromatin landscape. While the authors have identified SIRT1 as a key gene affected by the deficiency of IDH1, the mechanisms underlying IDH1's nuclear function are worth further exploration in future studies.

Overall, this study advances our understanding of the non-canonical localization of metabolic enzymes and their nuclear functions, shedding new light on their roles in cellular regulation.

---

## [Referee Report · Reviewer #3 (Public review)]

Li, Zhang, Wu and colleagues describe a new role for nuclear IDH1 in erythroid differentiation. IDH1 depletion results in a terminal erythroid differentiation defect with polychromatic and orthochromatic erythroblasts showing abnormal nuclei, nuclear condensation defects and an increased proportion of euchromatin, as well as enucleation defects. Using ChIP-seq for the histone modifications H3K79me3, H3K27me2 and H3K9me3, as well as ATAC-seq and RNA-seq in primary CD34-derived erythroblasts, the authors elucidate SIRT1 as a key dysregulated gene that is upregulated upon IDH1 knockdown. They furthermore show that chemical inhibition of SIRT1 partially rescues the abnormal nuclear morphology and enucleation defect during IDH1-deficient erythroid differentiation. The phenotype of delayed erythroid maturation and enucleation upon IDH1 shRNA-mediated knockdown was described in the group's previous co-authored study (PMID: 33535038). The authors describe this new role of IDH1 as non-canonical, but more experiments will be needed to determine whether this function of IDH1 in chromatin organization is secondary to its enzymatic-metabolic role. On the other hand, while the dependency of IDH1 mutant cells on NAD+ as well as a cell survival benefit upon SIRT1 inhibition has already been shown (see, e.g, PMID: 26678339, PMID: 32710757), previous studies focused on cancer cell lines and did not look at a developmental differentiation process, which makes this study interesting.

The authors had initially hypothesized that IDH1 has a role in the nucleus independent of its enzymatic function, which is interesting but was not supported by the presented experiments. In the revised manuscript, the authors decided to just focus on the nuclear role of IDH1. To this end, they present a system in HUDEP-2 cells harboring a CRISPR/Cas9-mediated IDH1 knockout and overexpression of an IDH1 construct containing a nuclear export signal. While they only use this system in some of their experiments, they mostly use a global IDH1 shRNA knockdown approach is employed, which will affect both forms of IDH1, cytoplasmic and nuclear. Future work using their system that specifically depletes nuclear IDH1 could further delineate changes of the chromatin landscape upon loss of nuclear IDH1 and also address how loss of nuclear IDH1 affects the part of the TCA cycle that has recently been shown to be present in the nucleus (PMID: 36044572).

---

## [Author Response]

The following is the authors’ response to the original reviews.

**Public Reviews:**

**Reviewer #1 (Public Review):**
The manuscript by Li et al. investigates the metabolism-independent role of nuclear IDH1 in chromatin state reprogramming during erythropoiesis. The authors describe accumulation and redistribution of histone H3K79me3, and downregulation of SIRT1, as a cause for dyserythropoiesis observed due to IDH1 deficiency. The authors studied the consequences of IDH1 knockdown, and targeted knockout of nuclear IDH1, in normal human erythroid cells derived from hematopoietic stem and progenitor cells and HUDEP2 cells respectively. They further correlate some of the observations such as nuclear localization of IDH1 and aberrant localization of histone modifications in MDS and AML patient samples harboring IDH1 mutations. These observations are intriguing from a mechanistic perspective and they hold therapeutic significance, however there are major concerns that make the inferences presented in the manuscript less convincing.(1) The authors show the presence of nuclear IDH1 both by cell fractionation and IF, and employ an efficient strategy to knock out nuclear IDH1 (knockout IDH1/ Sg-IDH1 and rescue with the NES tagged IDH1/ Sg-NES-IDH1 that does not enter the nucleus) in HUDEP2 cells. However, some important controls are missing.A) In Figure 3C, for IDH1 staining, Sg-IDH1 knockout control is missing.

Thanks for the reviewer’s suggestion. We have complemented the staining of Sg-IDH1 knockout cells, and made corresponding revision in Figure 3C in the revised manuscript.

B) Wild-type IDH1 rescue control (ie., IDH1 without NES tag) is missing to gauge the maximum rescue that is possible with this system.

Thanks for the reviewer’s suggestion. We have overexpressed wild-type IDH1 in the IDH1-deficient HUDEP2 cell line and detected the phenotype. The results are presented in Supplementary Figure 9 in the revised manuscript. As shown in Supplementary Figure 9A, IDH1 deficiency resulted in reduced cell number in HUDEP2 cells, a phenotype that was rescued by overexpression of wild-type IDH1 but not by NES-IDH1. Given IDH1's well-established role in redox homeostasis through catalyzing isocitrate to α-KG conversion, we hypothesized that both wild-type IDH1 and NES-IDH1 overexpression would significantly restore α-KG levels compared to the IDH1-deficient group. Supplementary Figure 9B demonstrates that IDH1 depletion resulted in a dramatic decrease in α-KG levels, whereas overexpression of either wild-type IDH1 or NES-IDH1 almost completely restored α-KG levels, as anticipated. These results suggest that wild-type IDH1 overexpression can restore metabolic regulatory functions as effectively as NES-IDH1 overexpression. To investigate whether apoptosis contributes to the impaired cell expansion caused by IDH1 deficiency, we performed Annexin V/PI staining to quantify apoptotic cells. As shown in Supplementary Figure 9C and D, flow cytometry analysis revealed no significant changes in apoptosis rates following either IDH1 depletion or ectopic expression of wild-type IDH1 or NES-IDH1 in IDH1 deficient HUDEP2 cells.

Flow cytometric analysis demonstrated that IDH1 deficiency triggered S-phase accumulation at day 8, indicative of cell cycle arrest. Whereas ectopic expression of wild-type IDH1 significantly rescued this cell cycle defect, overexpression of NES-IDH1 failed to ameliorate the S-phase accumulation phenotype induced by IDH1 depletion, as presented in Supplementary Figure 9E and F. Although NES-IDH1 overexpression rescued metabolic regulatory function defect, it failed to alleviate the cell cycle arrest induced by IDH1 deficiency. In contrast, wild-type IDH1 overexpression fully restored normal cell cycle progression. This functional dichotomy demonstrates that nuclear-localized IDH1 executes critical roles distinct from its cytoplasmic counterpart, and overexpression of wild-type IDH1 could efficient restore the functional impairment induced by depletion of nuclear localized IDH1.

(2) Considering the nuclear knockout of IDH1 (Sg-NES-IDH1 referenced in the previous point) is a key experimental system that the authors have employed to delineate non-metabolic functions of IDH1 in human erythropoiesis, some critical experiments are lacking to make convincing inferences.A) The authors rely on IF to show the nuclear deletion of Sg-NES-IDH1 HUDEP2 cells. As mentioned earlier since a knockout control is missing in IF experiments, a cellular fractionation experiment (similar to what is shown in Figure 2F) is required to convincingly show the nuclear deletion in these cells.

We sincerely thank the reviewer for raising this critical point. As suggested, we have performed additional IF experiments and cellular fractionation experiments to comprehensively address the subcellular localization of IDH1.

The results of IF staining were shown in Figure 3C of the revised manuscript. In Control HUDEP2 cells, endogenous IDH1 was detected in both the cytoplasm and nucleus. This dual localization may reflect its dynamic roles in cytoplasmic metabolic processes and potential nuclear functions under specific conditions. In Sg-IDH1 cells (IDH1 knockout), IDH1 signal was undetectable, confirming efficient depletion of the protein. In Sg-NES-IDH1 cells (overexpressing NES-IDH1 in IDH1 deficient cells), IDH1 predominantly accumulated in the cytoplasm, consistent with the disruption of its nuclear export signal. The dual localization of IDH1 that was determined by IF staining experiment were then further confirmed by cellular fractionation assays, as shown in Figure 3D.

B) Since the authors attribute nuclear localization to a lack of metabolic/enzymatic functions, it is important to show the status of ROS and alpha-KG in the Sg-NES-IDH1 in comparison to control, wild type rescue, and knockout HUDEP2 cells. The authors observe an increase of ROS and a decrease of alpha-KG upon IDH1 knockdown. If nuclear IDH1 is not involved in metabolic functions, is there only a minimal or no impact of the nuclear knockout of IDH1 on ROS and alpha-KG, in comparison to complete knockout? These studies are lacking.

We appreciate the insightful suggestions of the reviewers and agree that the detection of ROS and alpha-KG is useful for the demonstration of the non-canonical function of IDH1. We examined alpha-KG concentrations in control, IDH1 knockout and nuclear IDH1 knockout HUDEP2 cell lines. The results showed a significant decrease in alpha-KG content after complete knockout of IDH1, whereas there was no significant change in nuclear knockout IDH1 (Supplementary Figure 9B). As to the detection of ROS level, the commercial ROS assay kits that we can get are detected using PE (Excitation: 565nm; Emission: 575nm) and FITC (Excitation: 488nm; Emission: 518nm) channels in flow cytometry. We constructed HUDEP2 cell lines of Sg-IDH1 and Sg-NES-IDH1 to express green fluorescent protein (Excitation: 488nm; Emission: 507nm) and Kusabira Orange fluorescent protein (Excitation: 500nm; Emission: 561nm) by themselves. Unfortunately, due to the spectral overlap of the fluorescence channels, we were unable to detect the changes in ROS levels in these HUDEP2 cell lines using the available commercial kit.

(3) The authors report abnormal nuclear phenotype in IDH1 deficient erythroid cells. It is not clear what parameters are used here to define and quantify abnormal nuclei. Based on the cytospins (eg., Figure 1A, 3D) many multinucleated cells are seen in both shIDH1 and Sg-NES-IDH1 erythroid cells, compared to control cells. Importantly, this phenotype and enucleation defects are not rescued by the administration of alpha-KG (Figures 1E, F). The authors study these nuclei with electron microscopy and report increased euchromatin in Figure 4B. However, there is no discussion or quantification of polyploidy/multinucleation in the IDH1 deficient cells, despite their increased presence in the cytospins.A) PI staining followed by cell cycle FACS will be helpful in gauging the extent of polyploidy in IDH1 deficient cells and could add to the discussions of the defects related to abnormal nuclei.

We appreciate the reviewer’s insightful suggestion. Since PI dye is detected using the PE channel (Excitation: 565nm; Emission: 575nm) of the flow cytometer and the HUDEP2 cell line expresses Kusabira orange fluorescent protein (Excitation: 500nm; Emission: 561nm), we were unable to use PI staining to detect the cell cycle. Edu staining is another commonly used method to determine cell cycle progression, and we performed Edu staining followed by flow cytometry analysis on Control, Sg-IDH1 and Sg-NES-IDH1 HUDEP2 cells, respectively. The results showed that complete knockdown of IDH1 resulted in S-phase block and increased polyploidy in HUDEP2 cells on day 8 of erythroid differentiation, and overexpression of IDH1-NES did not reverse this phenotype (Supplemental Figure 9E-F). Moreover, we have added a discussion of abnormal nuclei being associated with impaired erythropoiesis.

B) For electron microscopy quantification in Figures 4B and C, how the quantification was done and the labelling of the y-axis (% of euchromatin and heterochromatin) in Figure 4 C is not clear and is confusingly presented. The details on how the quantification was done and a clear label (y-axis in Figure 4C) for the quantification are needed.

Thanks for the reviewer's suggestion. In this study, we calculated the area of nuclear, heterochromatin and euchromatin by using Image J software. We addressed the quantification strategy in the section of Supplementary methods of the revised Supplementary file. In addition, the y-axis label in Figure 4C was changed to “the area percentage of euchromatin and heterochromatin’’.

C) As mentioned earlier, what parameters were used to define and quantify abnormal nuclei (e.g. Figure 1A) needs to be discussed clearly. The red arrows in Figure 1A all point to bi/multinucleated cells. If this is the case, this needs to be made clear.

We thank the reviewer for their suggestion. In our present study, nuclear malformations were defined as cells exhibiting binucleation or multinucleation based on cytospin analysis. A minimum of 300 cells per group were evaluated, and the proportion of aberrant nuclei was calculated as (number of abnormal cells / total counted cells) × 100%.

(4) The authors mention that their previous study (reference #22) showed that ROS scavengers did not rescue dyseythropoiesis in shIDH1 cells. However, in this referenced study they did report that vitamin C, a ROS scavenger, partially rescued enucleation in IDH1 deficient cells and completely suppressed abnormal nuclei in both control and IDH1 deficient cells, in addition to restoring redox homeostasis by scavenging reactive oxygen species in shIDH1 erythroid cells. In the current study, the authors used ROS scavengers GSH and NAC in shIDH1 erythroid cells and showed that they do not rescue abnormal nuclei phenotype and enucleation defects. The differences between the results in their previous study with vitamin C vs GSH and NAC in the context of IDH1 deficiency need to be discussed.

We appreciate the reviewer’s insightful observation. The apparent discrepancy between the effects of vitamin C (VC) in our previous study and glutathione (GSH)/N-acetylcysteine (NAC) in the current work can be attributed to divergent molecular mechanisms beyond ROS scavenging. A growing body of evidence has identified vitamin C as a multifunctional regulator. In addition to acting as an antioxidant maintaining redox homeostasis, VC also acts as a critical epigenetic modulator. VC have been identified as a cofactor for α-ketoglutarate (α-KG)-dependent dioxygenases, including TET2, which catalyzes 5-methylcytosine (5mC) oxidation to 5-hydroxymethylcytosine (5hmC) [1,2]. Structural studies confirm its direct interaction with TET2’s catalytic domain to enhance enzymatic activity in vitro [3]. The biological significance of the epigenetic modulation induced by vitamin C is illustrated by its ability to improve the generation of induced pluripotent stem cells and to induce a blastocyst-like state in mouse embryonic stem cells by promoting demethylation of H3K9 and 5mC, respectively [4,5]. In contrast, GSH and NAC are canonical ROS scavengers lacking intrinsic epigenetic-modifying activity. While they effectively neutralize oxidative stress (as validated by reduced ROS levels in our current data, Supplemental Figure 7), their inability to rescue nuclear abnormalities or enucleation defects in IDH1 deficient cells suggests that IDH1 deficiency-driven dyserythropoiesis is not solely ROS-dependent.

References:

(1) Blaschke K, Ebata KT, Karimi MM, Zepeda-Martínez JA, Goyal P, et al. Vitamin C induces Tet-dependent DNA demethylation and a blastocyst-like state in ES cells. Nature. 20138;500(7461): 222-226.

(2) Minor EA, Court BL, Young JI, Wang G. Ascorbate induces ten-eleven translocation (Tet) methylcytosine dioxygenase-mediated generation of 5-hydroxymethylcytosine. J Biol Chem. 2013;288(19): 13669-13674.

(3) Yin R, Mao S, Zhao B, Chong Z, Yang Y, et al. Ascorbic acid enhances Tet-mediated 5-methylcytosine oxidation and promotes DNA demethylation in mammals. J Am Chem Soc. 2013;135(28):10396-10403.

(4) Esteban MA, Wang T, Qin B, Yang J, Qin D, et al. Vitamin C enhances the generation of mouse and human induced pluripotent stem cells. Cell Stem Cell. 2010;6(1):71-79.

(5) Chung T, Brena RM, Kolle G, Grimmond SM, Berman BP, et al. Vitamin C promotes widespread yet specific DNA demethylation of the epigenome in human embryonic stem cells. Stem Cells. 2010;28(10):1848-1855.

(5) The authors describe an increase in euchromatin as the consequential abnormal nuclei phenotype in shIDH1 erythroid cells. However, in their RNA-seq, they observe an almost equal number of genes that are up and down-regulated in shIDH1 cells compared to control cells. If possible, an RNA-Seq in nuclear knockout Sg-NES-IDH1 erythroid cells in comparison with knockout and wild-type cells will be helpful to tease out whether a specific absence of IDH1 in the nucleus (ie., lack of metabolic functions of IDH) impacts gene expression differently.

Thanks for the reviewer's suggestion. ATAC-seq showed an increase in chromatin accessibility after IDH1 deletion, but the number of up-regulated genes was slightly larger than that of down-regulated genes, which may be caused by the metabolic changes affected by IDH1 deletion. In order to explore the effect of chromatin accessibility changes on gene expression after IDH1 deletion, we analyzed the changes in differential gene expression at the differential ATAC peak region (as shown in Author response image 1), and the results showed that the gene expression at the ATAC peak region with increased chromatin accessibility was significantly up-regulated. This may explain the regulation of chromatin accessibility on gene expression.

**Author response image 1. sa4fig1:** Box plots of gene expression differences of differential ATAC peaks located in promoter for the signal increasing and decreasing groups.

(6) In Figure 8, the authors show data related to SIRT1's role in mediating non-metabolic, chromatin-associated functions of IDH1.A) The authors show that SIRT1 inhibition leads to a rescue of enucleation and abnormal nuclei. However, whether this rescues the progression through the late stages of terminal differentiation and the euchromatin/heterochromatin ratio is not clear.

Thanks for the reviewer's suggestion. As shown in Supplementary Figure 14 and 15 in the revised Supplementary Data, our data showed that both the treatment of SRT1720 on normal erythroid cells and treatment of IDH1-deficient erythroid cells with SIRT1 inhibitor both have no effect on the terminal differentiation.

(7) In Figure 4 and Supplemental Figure 8, the authors show the accumulation and altered cellular localization of H3K79me3, H3K9me3, and H3K27me2, and the lack of accumulation of other three histone modifications they tested (H3K4me3, H3K35me4, and H3K36me2) in shIDH1 cells. They also show the accumulation and altered localization of the specific histone marks in Sg-NES-IDH1 HUDEP2 cells.A) To aid better comparison of these histone modifications, it will be helpful to show the cell fractionation data of the three histone modifications that did not accumulate (H3K4me3, H3K35me4, and H3K36me2), similar to what was shown in Figure 4E for H3K79me3, H3K9me3, and H3K27me2).

We appreciate the reviewer’s insightful suggestion. We collected erythroblasts on day 15 of differentiation from cord blood-derived CD34^+^ hematopoietic stem cells to erythroid lineage and performed ChIP assay. As shown in Author response image 2, the results showed that the concentration of bound DNA of H3K9me3, H3K27me2 and H3K79me3 was too low to meet the sequencing quality requirement, which was consistent with that of WB. In addition, we tried to perform ChIP-seq for H3K79me3, and the results showed that there was no marked peak signal.

**Author response image 2. sa4fig2:** ChIP-seq analysis show that there was no marked peak signal of H3K79me3 on D15. (A) Quality control of ChIP assay for H3K9me3, H3K27me2, and H3K79me3. (B) Representative peaks chart image showed normalized ChIP signal of H3K79me3 at gene body regions. (C) Heatmaps displayed normalized ChIP signal of H3K79me3 at gene body regions. The window represents ±1.5 kb regions from the gene body. TES, transcriptional end site; TSS, transcriptional start site.

C) Among the three histone marks that are dysregulated in IDH1 deficient cells (H3K79me3, H3K9me3, and H3K27me2), the authors show via ChIP-seq (Fig5) that H3K79me3 is the critical factor. However, the ChIP-seq data shown here lacks many details and this makes it hard to interpret the data. For example, in Figure 5A, they do not mention which samples the data shown correspond to (are these differential peaks in shIDH1 compared to shLuc cells?). There is also no mention of how many replicates were used for the ChIP seq studies.

We thank the reviewer for pointing this out. We apologize for not clearly describing the ChIP-seq data for H3K9me3, H3K27me2 and H3K79me3 and we have revised them in the corresponding paragraphs. Since H3 proteins gradually translocate from the nucleus to the cytoplasm starting at day 11 (late Baso-E/Ploy-E) of erythroid lineage differentiation, we performed ChIP-seq for H3K9me3, H3K27me2 and H3K79me3 only for the shIDH1 group, and set up three independent biological replicates for each of them.

**Reviewer #2 (Public Review):**
Li and colleagues investigate the enzymatic activity-independent function of IDH1 in regulating erythropoiesis. This manuscript reveals that IDH1 deficiency in the nucleus leads to the redistribution of histone marks (especially H3K79me3) and chromatin state reprogramming. Their findings suggest a non-typical localization and function of the metabolic enzyme, providing new insights for further studies into the non-metabolic roles of metabolic enzymes. However, there are still some issues that need addressing:(1) Could the authors show the RNA and protein expression levels (without fractionation) of IDH1 on different days throughout the human CD34+ erythroid differentiation?

We sincerely appreciate the reviewer’s constructive feedback. To address this point, we have now systematically quantified IDH1 expression dynamics across erythropoiesis stages using qRT-PCR and Western blot analyses. As quantified in Supplementary fige 1, IDH1 expression exhibited a progressive upregulation during early erythropoiesis and subsequently stabilized throughout terminal differentiation.

(2) Even though the human CD34+ erythroid differentiation protocol was published and cited in the manuscript, it would be helpful to specify which erythroid stages correspond to cells on days 7, 9, 11, 13, and 15.

We sincerely thank the reviewer for raising this important methodological consideration. Our research group has previously established a robust platform for staged human erythropoiesis characterization using cord blood-derived CD34^+^ hematopoietic stem cells (HSCs) [6-9]. This standardized protocol enables high-purity isolation and functional analysis of erythroblasts at defined differentiation stages.

Thanks for the reviewer’s suggestion. Our previous work (Jingping Hu et.al, Blood 2013. Xu Han et.al, Blood 2017.Yaomei Wang et.al, Blood 2021.) have isolation and functional characterization of human erythroblasts at distinct stages by using Cord blood. These works illustrated that using cord blood-derived hematopoietic stem cells and purification each stage of human erythrocytes can facilitate a comprehensive cellular and molecular characterization.

Following isolation from cord blood, CD34^+^ cells were cultured in a serum-free medium and induced to undergo erythroid differentiation using our standardized protocol. The process of erythropoiesis was comprised of 2 phases. During the early phase (day 0 to day 6), hematopoietic stem progenitor cells expanded and differentiated into erythroid progenitors, including BFU-E and CFU-E cells.

During terminal erythroid maturation (day 7 to day 15), CFU-E cells progressively transitioned through defined erythroblast stages, as validated by daily cytospin morphology and expression of band 3/α4 integrin. The stage-specific composition was quantitatively characterized as follows:

**Author response table 1. sa4table1:** The composition of erythroblast during terminal stage erythropoiesis.

Day	Pro-E	Baso-E	Poly-E	Ortho-E
7	50%	50%	\	\
9	10%	80%	10%	\
11	\	65%	20%	15%
13	\	20%	30%	50%
15	\	10%	20%	70%

This differentiation progression from proerythroblasts (Pro-E) through basophilic (Baso-E), polychromatic (Poly-E), to orthochromatic erythroblasts (Ortho-E) recapitulates physiological human erythropoiesis, confirming the validity of our differentiation system for mechanistic studies.

Reference:

(6) Li J, Hale J, Bhagia P, Xue F, Chen L, et al. Isolation and transcriptome analyses of human erythroid progenitors: BFU-E and CFU-E. Blood. 2014;124(24):3636-3645.

(7) Hu J, Liu J, Xue F, Halverson G, Reid M, et al. Isolation and functional characterization of human erythroblasts at distinct stages: implications for understanding of normal and disordered erythropoiesis in vivo. Blood. 2013;121(16):3246-3253.

(8) Wang Y, Li W, Schulz VP, Zhao H, Qu X, et al. Impairment of human terminal erythroid differentiation by histone deacetylase 5 deficiency. Blood. 2021;138(17):1615-1627.

(9) Li M, Liu D, Xue F, Zhang H, Yang Q, et al. Stage-specific dual function: EZH2 regulates human erythropoiesis by eliciting histone and non-histone methylation. Haematologica. 2023;108(9):2487-2502.

(3) It is important to mention on which day the lentiviral knockdown of IDH1 was performed. Will the phenotype differ if the knockdown is performed in early vs. late erythropoiesis? In Figures 1C and 1D, on which day do the authors begin the knockdown of IDH1 and administer NAC and GSH treatments?

We sincerely appreciate the reviewer’s inquiry regarding the experimental timeline. The day of getting CD34^+^ cells was recorded as day 0. Lentivirus of IDH1-shRNA and Luciferase -shRNA was transduced in human CD34^+^ at day 2. Puromycin selection was initiated 24h post-transduction to eliminate non-transduced cells. IDH1-KD cells were then selected for 3 days. The knock down deficiency of IDH1 was determined on day 7. NAC or GSH was added to culture medium and replenished every 2 days.

(4) While the cell phenotype of IDH1 deficiency is quite dramatic, yielding cells with larger nuclei and multi-nuclei, the authors only attribute this phenotype to defects in chromatin condensation. Is it possible that IDH1-knockdown cells also exhibit primary defects in mitosis/cytokinesis (not just secondary to the nuclear condensation defect ?), given the function of H3K79Me in cell cycle regulation?

We appreciate the reviewer’s insightful suggestion. We performed Edu based cell cycle analysis on Control, Sg-IDH1 and Sg-NES-IDH1 HUDEP2 cells, respectively. The results showed that IDH1 deficiency resulted in S-phase block and increased polyploidy in HUDEP2 cells on day 8 of erythroid differentiation. NES-IDH1 overexpression failed to rescue these defects, indicating nuclear IDH1 depletion as the primary driving factor (Figure 3E,F). Recent studies have established a clear link between cell cycle arrest and nuclear malformation. Chromosome mis-segregation, nuclear lamina disruption, mechanical stress on the nuclear envelope, and nucleolar dysfunction all contribute to nuclear abnormalities that trigger cell cycle checkpoints [10,11]. Based on all these findings, it reasonable for us to speculate that the cell cycle defect in IDH1 deficient cells might caused by the nuclear malfunction.

Reference:

(10) Hong T, Hogger AC, Wang D, Pan Q, Gansel J, et al. Cell Death Discov. CDK4/6 inhibition initiates cell cycle arrest by nuclear translocation of RB and induces a multistep molecular response. 2024;10(1):453.

(11) Hervé S, Scelfo A, Marchisio GB, Grison M, Vaidžiulytė K, et al. Chromosome mis-segregation triggers cell cycle arrest through a mechanosensitive nuclear envelope checkpoint. Nat Cell Biol. 2025;27(1):73-86.

(5) Why are there two bands of Histone H3 in Figure 4A?

We sincerely appreciate the reviewer's insightful observation regarding the dual bands of Histone H3 in our original Figure 4A. Upon rigorous investigation, we identified that the observed doublet pattern likely originated from the inter-batch variability of the original commercial antibody. To conclusively resolve this technical artifact, we have procured a new lot of Histone H3 antibody and repeated the western blot experimental under optimized conditions, and the results demonstrates a single band of H3.

(6) Displaying a heatmap and profile plots in Figure 5A between control and IDH1-deficient cells will help illustrate changes in H3K79me3 density in the nucleus after IDH1 knockdown.

Thank you for your suggestion. As presented in Author response image 2, we performed ChIP assays on erythroblasts collected at day 15. However, the concentration of H3K79me3-bound DNA was insufficient to meet the quality threshold required for reliable sequencing. Consequently, we are unable to generate the requested heatmap and profile plots due to limitations in data integrity from this experimental condition.

**Reviewer #3 (Public Review):**
Li, Zhang, Wu, and colleagues describe a new role for nuclear IDH1 in erythroid differentiation independent from its enzymatic function. IDH1 depletion results in a terminal erythroid differentiation defect with polychromatic and orthochromatic erythroblasts showing abnormal nuclei, nuclear condensation defects, and an increased proportion of euchromatin, as well as enucleation defects. Using ChIP-seq for the histone modifications H3K79me3, H3K27me2, and H3K9me3, as well as ATAC-seq and RNA-seq in primary CD34-derived erythroblasts, the authors elucidate SIRT1 as a key dysregulated gene that is upregulated upon IDH1 knockdown. They furthermore show that chemical inhibition of SIRT1 partially rescues the abnormal nuclear morphology and enucleation defect during IDH1-deficient erythroid differentiation. The phenotype of delayed erythroid maturation and enucleation upon IDH1 shRNA-mediated knockdown was described in the group's previous co-authored study (PMID: 33535038). The authors' new hypothesis of an enzyme- and metabolism-independent role of IDH1 in this process is currently not supported by conclusive experimental evidence as discussed in more detail further below. On the other hand, while the dependency of IDH1 mutant cells on NAD+, as well as cell survival benefit upon SIRT1 inhibition, has already been shown (see, e.g, PMID: 26678339, PMID: 32710757), previous studies focused on cancer cell lines and did not look at a developmental differentiation process, which makes this study interesting.(1) The central hypothesis that IDH1 has a role independent of its enzymatic function is interesting but not supported by the experiments. One of the author's supporting arguments for their claim is that alpha-ketoglutarate (aKG) does not rescue the IDH1 phenotype of reduced enucleation. However, in the group's previous co-authored study (PMID: 33535038), they show that when IDH1 is knocked down, the addition of aKG even exacerbates the reduced enucleation phenotype, which could indicate that aKG catalysis by cytoplasmic IDH1 enzyme is important during terminal erythroid differentiation. A definitive experiment to test the requirement of IDH1's enzymatic function in erythropoiesis would be to knock down/out IDH1 and re-express an IDH1 catalytic site mutant. The authors perform an interesting genetic manipulation in HUDEP-2 cells to address a nucleus-specific role of IDH1 through CRISPR/Cas-mediated IDH1 knockout followed by overexpression of an IDH1 construct containing a nuclear export signal. However, this system is only used to show nuclear abnormalities and (not quantified) accumulation of H3K79me3 upon nuclear exclusion of IDH1. Otherwise, a global IDH1 shRNA knockdown approach is employed, which will affect both forms of IDH1, cytoplasmic and nuclear. In this system and even the NES-IDH1 system, an enzymatic role of IDH1 cannot be excluded because (1) shRNA selection takes several days, prohibiting the assessment of direct effects of IDH1 loss of function (only a degron approach could address this if IDH1's half-life is short), and (2) metabolic activity of this part of the TCA cycle in the nucleus has recently been demonstrated (PMID: 36044572), and thus even a nuclear role of IDH1 could be linked to its enzymatic function, which makes it a challenging task to separate two functions if they exist.

We appreciate the reviewer’s emphasis on rigorously distinguishing between enzymatic and enzymatic independent roles of IDH1. In our revised manuscript, we have removed all assertions of a "metabolism-independent" mechanism. Instead, we focus on demonstrating that nuclear-localized IDH1 contributes to chromatin state regulation during terminal erythropoiesis (e.g., H3K79me3 accumulation). While we acknowledge that nuclear IDH1’s enzymatic activity may still play a role [12], our data emphasize its spatial association with chromatin remodeling. We now explicitly state that nuclear IDH1’s function may involve both enzymatic and structural roles, and further studies are required to dissect these mechanisms.

Reference:

(12) Kafkia E, Andres-Pons A, Ganter K, Seiler M, Smith TS, et al.Operation of a TCA cycle subnetwork in the mammalian nucleus. Sci Adv. 2022;8(35):eabq5206.

(2) It is not clear how the enrichment of H3K9me3, a prominent marker of heterochromatin, upon IDH1 knockdown in the primary erythroid culture (Figure 4), goes along with a 2-3-fold increase in euchromatin. Furthermore, in the immunofluorescence (IF) experiments presented in Figure 4Db, it seems that H3K9me3 levels decrease in intensity (the signal seems more diffuse), which seems to contrast the ChIP-seq data. It would be interesting to test for localization of other heterochromatin marks such as HP1gamma. As a related point, it is not clear at what stage of erythroid differentiation the ATAC-seq was performed upon luciferase- and IDH1-shRNA-mediated knockdown shown in Figure 6. If it was done at a similar stage (Day 15) as the electron microscopy in Figure 4B, then the authors should explain the discrepancy between the vast increase in euchromatin and the rather small increase in ATAC-seq signal upon IDH1 knockdown.

Thank you for raising this important point. We agree that while H3K9me3 and H3K27me2 modifications are detectable in the nucleus, their functional association with chromatin in this context remains unclear. Our ChIP-seq data did not reveal distinct enrichment peaks for H3K9me3 or H3K27me2 (unlike the well-defined H3K79me3 peaks), suggesting that these marks may not be stably bound to specific chromatin regions under the experimental conditions tested. However, we acknowledge that the absence of clear peaks in our dataset does not definitively rule out chromatin interactions, as technical limitations or transient binding dynamics could influence these results. To avoid over-interpretation, we have removed speculative statements about the chromatin-unbound status of H3K9me3 and H3K27me2 from the revised manuscript. This revision aligns with our broader effort to present conclusions strictly supported by the current data while highlighting open questions for future investigation.

(3)The subcellular localization of IDH1, in particular its presence on chromatin, is not convincing in light of histone H3 being enriched in the cytoplasm on the same Western blot. H3 would be expected to be mostly localized to the chromatin fraction (see, e.g., PMID: 31408165 that the authors cite). The same issue is seen in Figure 4A.

We sincerely appreciate the reviewer's insightful comment regarding the subcellular distribution of histone H3 in our study. We agree that histone H3 is classically associated with chromatin-bound fractions, and its cytoplasmic enrichment in our Western blot analyses appears counterintuitive at first glance. However, this observation is fully consistent with the unique biology of terminal erythroid differentiation, which involves drastic nuclear remodeling and histone release - a hallmark of terminal stage erythropoiesis. Terminal erythroid differentiation is characterized by progressive nuclear condensation, chromatin compaction, and eventual enucleation. During this phase, global chromatin reorganization leads to the active eviction of histones from the condensed nucleus into the cytoplasm. This process has been extensively documented in erythroid cells, with studies demonstrating cytoplasmic accumulation of histones H3 and H4 as a direct consequence of nuclear envelope breakdown and chromatin decondensation preceding enucleation [13-16]. Our experiments specifically analyzed terminal-stage polychromatic and orthochromatic erythroblasts. At this stage, histone releasing into the cytoplasm is a dominant biological event, explaining the pronounced cytoplasmic H3 signal in our subcellular fractionation assays.

In summary, the cytoplasmic enrichment of histone H3 in our data aligns with established principles of erythroid biology and reinforces the physiological relevance of our findings. We thank the reviewer for raising this critical point, which allowed us to better articulate the unique aspects of our experimental system.

Reference:

(13) Hattangadi SM, Martinez-Morilla S, Patterson HC, Shi J, Burke K, et al. Histones to the cytosol: exportin 7 is essential for normal terminal erythroid nuclear maturation. Blood. 2014;124(12):1931-1940.

(14) Zhao B, Mei Y, Schipma MJ, Roth EW, Bleher R, et al. Nuclear Condensation during Mouse Erythropoiesis Requires Caspase-3-Mediated Nuclear Opening. Dev Cell. 2016;36(5): 498-510.

(15) Zhao B, Liu H, Mei Y, Liu Y, Han X, et al. Disruption of erythroid nuclear opening and histone release in myelodysplastic syndromes. Cancer Med. 2019;8(3):1169-1174.

(16) Zhen R, Moo C, Zhao Z, Chen M, Feng H, et al. Wdr26 regulates nuclear condensation in developing erythroblasts. Blood. 2020;135(3):208-219.

(4) This manuscript will highly benefit from more precise and complete explanations of the experiments performed, the material and methods used, and the results presented. At times, the wording is confusing. As an example, one of the "Key points" is described as "Dyserythropoiesis is caused by downregulation of SIRT1 induced by H3K79me3 accumulation." It should probably read "upregulation of SIRT1".

We sincerely thank the reviewer for highlighting the need for improved clarity in our experimental descriptions and textual precision. We fully agree that rigorous wording is essential to accurately convey scientific findings. Specific modifications have been made and are highlighted in Track Changes mode in the resubmitted manuscript.

The reviewer correctly identified an inconsistency in the original phrasing of one key finding. The sentence in question ("Dyserythropoiesis is caused by downregulation of SIRT1 induced by H3K79me3 accumulation") has been revised to:"Dyserythropoiesis is caused by the upregulation of SIRT1 mediated through H3K79me3 accumulation." This correction aligns with our experimental data showing that H3K79me3 elevation promotes SIRT1 transcriptional activation. We apologize for this oversight and have verified the consistency of all regulatory claims in the text.

**Recommendations for the authors:**

**Reviewer #1 (Recommendations For The Authors):**
(1) It will be helpful to mention/introduce the cells used for the study at the beginning of the results section. For example, for Figure 1A neither the figure legend nor the results text includes information on the cells used.

Thanks for the reviewer’s suggestion. The detail information of the cells that were used in our study have been provided in the revised manuscript.

(2) Important details for many figures are lacking. For example, in Figure 5, there is no mention of the replicates for ChIP-Seq studies. Also, the criteria used for quantifications of abnormal nuclei, % euchromatin vs heterochromatin, the numbers of biological replicates, and how many fields/cells were used for these quantifications are missing.

We thank the reviewer for emphasizing the importance of methodological transparency. It has been revised accordingly. The ChIP-Seq data in Figure 5 was generated from three independent biological replicates to ensure reproducibility. In this study, Image J software was used to calculate the area of nuclear, heterochromatin/euchromatin and to quantify the percentage of euchromatin and heterochromatin. A minimum of 300 cells per group were evaluated, and the proportion of aberrant nuclei was calculated as (number of abnormal cells / total counted cells) × 100%.

(3) It will be helpful if supplemental data are ordered according to how they are discussed in the text. Currently, the order of the supplemental data is hard to keep track of eg., the results section starts describing supplemental Figure 1, then the text jumps to supplemental Figure 5 followed by Supplemental Figure 3 (and so on).

Thanks for the reviewer’s suggestion. It has been revised accordingly.

(4) Overall, there are many incomplete sentences and typos throughout the manuscript including some of the figures e.g. on page 10 the sentence "Since the generation of erythroid with abnormal nucleus and reduction of mature red blood cells caused by IDH1 absence are notable characteristics of MDS and AML." is incomplete. On page 11, it reads "Histone post-modifications". This needs to be either histone modifications or histone post-translational modifications. In Figure 4C, the y-axis title is hard to understand "% of euchromatin and heterochromatin". Overall, the document needs to be proofread and revised carefully.

Thanks for the reviewer’s suggestion. We have made revision accordingly in the revised manuscript. The sentence "Since the generation of erythroid with abnormal nucleus and reduction of mature red blood cells caused by IDH1 absence are notable characteristics of MDS and AML." has been revised to “The production of erythrocytes with abnormal nuclei and the reduction of mature erythrocytes due to IDH1 deletion are prominent features of MDS and AML.” “% of euchromatin and heterochromatin” has been modified to “Area ratio of euchromatin to heterochromatin”.

**Reviewer #3 (Recommendations For The Authors):**
The following critique points aim to help the authors to improve their manuscript:(1) The authors reason (p. 10) that because mutant IDH1 has been shown to result in altered chromatin organization, this could be the case in their system, too. However, mutant IDH1 has an ascribed metabolic consequence, the generation of 2-HG, which further weakens the author's argument for an enzymatically independent role of IDH1 in their system. The same is true for the author's observation in Supplementary Figure 9B that in IDH1-mutant AML/MDS samples, H3K79me3 colocalized with the IDH1 mutants in the nucleus. Again, this speaks in favor of IDH1's role being linked to metabolism. The authors could re-write this manuscript, not so much emphasizing the separation of function between different subcellular forms of IDH1 but rather focusing on the chromatin changes and how they could be linked to the actual phenotype, the nuclear condensation and enucleation defect - if so, addressing the surprising finding of enrichment of both active and repressive chromatin marks will be important.

Thanks for the reviewer’s suggestion. We agree with the reviewers and editors all the data we present in the current are not robust enough to rigorously distinguish between enzymatic and enzymatic-independent roles of IDH1. In our revised manuscript, we have removed all assertions of a "metabolism-independent" mechanism. Instead, we focus on demonstrating that nuclear-localized IDH1 contributes to chromatin state regulation during terminal erythropoiesis (e.g., H3K79me3 accumulation).

(2) How come so many genes were downregulated by RNA-seq (about an equal number as upregulated genes) but not more open by ATAC-seq? The authors should discuss this result.

Thanks for the reviewer's suggestion. ATAC-seq showed an increase in chromatin accessibility after IDH1 deletion, but the number of up-regulated genes was slightly larger than that of down-regulated genes, which may be caused by the metabolic changes affected by IDH1 deletion. In order to explore the effect of chromatin accessibility changes on gene expression after IDH1 deletion, we analyzed the changes in differential gene expression at the differential ATAC peak region (as shown in the figure below), and the results showed that the gene expression at the ATAC peak region with increased chromatin accessibility was significantly up-regulated. This may explain the regulation of chromatin accessibility on gene expression.

(3) For the ChIP-seq analyses of H3K79me3, H3K27me2, and H3K9me3, the authors should not just show genome-wide data but also several example gene tracks to demonstrate the differential abundance of peaks in control versus IDH1 knockdown. Furthermore, the heatmap shown in Figure 5A should include broader regions spanning the gene bodies, to visualize the intergenic H3K27me2 and H3K9me3 peaks. Expression could very well be regulated from these intergenic regions as they could bear enhancer regions. ChIP-seq for H3K27Ac in the same setting would be very useful to identify those enhancers.

Thanks for the reviewer’s suggestion. It has been revised accordingly. We reanalyzed the ChIP-seq peak signal of H3K79me3, H3K27me2 and H3K9me3 in a wider region (±5Kb) at gene body, and the results showed that the H3K27me2 and H3K9me3 peak signals did not change significantly. Since H3K79me3 showed a higher peak signal and was mainly enriched in the promoter region, our subsequent analysis focusing on the impact of H3K79me3 accumulation on chromatin accessibility and gene expression might be more valuable.

**Author response image 3. sa4fig3:** ChIP-seq analysis show that the peak signal of H3K79me3,H3K27me2 and H3K9me3. (A) Heatmaps displayed normalized ChIP signal of H3K9me3, H3K27me2, and H3K79me3 at gene body regions. The window represents ±5 kb regions from the gene body. TES, transcriptional end site; TSS, transcriptional start site. (B) Representative peaks chart image showed normalized ChIP signal of H3K9me3, H3K27me2, and H3K79me3 at gene body regions.

(4) The absent or very mild delay (also no significance visible in the quantification plots) in the generation of orthochromatic erythroblasts on Day 13 upon IDH1 shRNA knockdown as per a4-integrin/Band3 flow cytometry does not correspond to the already quite prominent number of multinucleated cells at that stage seen by cytospin/Giemsa staining. Why do the authors think this is the case? Cytospin/Giemsa staining might be the better method to quantify this phenotype and the authors should quantify the cells at different stages in at least 100 cells from non-overlapping cytospin images.

Thanks for the reviewer’s suggestion. We have supplemented the cytpspin assay and the results were presented in Supplemental Figure 4.

(5) The pull-down assay in Figure 7E does not show a specific binding of H3K79me3 to the SIRT1 promoter. Rather, there is just more H3K79me3 in the nucleus, thus leading to generally increased binding. The authors should show that H3K79me3 does not bind more just everywhere but to specific loci. The ChIP-seq data mention only categories but don't show any gene lists that could hint at the specificity of H3K79me3 binding at genes that would promote nuclear abnormalities and enucleation defects.

We thank the reviewer for pointing this out. The GSEA results of H3K79me3 peak showed enrichment of chromatin related biological processes, and the list of associated genes is shown Figure 7B. In addition, we also displayed the changes in H3K79me3 peak signals, ATAC peak signals, and gene expression at gene loci of three chromatin-associated genes (SIRT1, KMT5A and NUCKS1).

(6) P. 12: "Representatively, gene expression levels and ATAC peak signals at SIRT1 locus were elevated in IDH1-shRNA group and were accompanied by enrichment of H3K9me3 (Figure 7F)." Figure 7F does not show an enrichment of H3K9me3, but if the authors found such, they should explain how this modification correlates with the activation of gene expression.

Thank you for bringing this issue to our attention. We sincerely apologize for the mistake in the description of Figure 7F on page 12. We have already corrected this error in the revised manuscript.

(7) Related to the mild phenotype by flow cytometry on Day 13, are the "3 independent biological replicates" from culturing and differentiating CD34 cells from 3 different donors? If all are from the same donor, experiments from at least a second donor should be performed to generalize the results.

In our current study, CD34^+^ cells were derived from different donors.

(8) If the images in Supplementary Figure 4 are only the indicated cell type, then it is not clear how the data were quantified since only some cells in each image are pointed at and others do not seem to have as large nuclei. There is also no explanation in the legend what the colors mean (nuclei were presumably stained with DAPI, not clear what the cytoplasm stain is - GPA?).

We thank the reviewer for pointing this out. We have revised the manuscript accordingly. Specifically, the nuclei was stained with DAPI and the color was blue. The cell membrane was stained with GPA and the color was red. This staining method allows for clear visualization of the cell structure and helps to better understand the localization of the proteins of interest.

(9) It is not clear to this reviewer whether Figure 4F is a quantification of the Western Blot or of the IF data.

Figure 4F is a quantification of the Western Blot experiment.

(10) The authors sometimes do not describe experiments well, e.g., "treatment of IDH1-deficient erythroid cells with IDH1-EX527" (p. 13). EX-527 is a SIRT1 inhibitor, which the authors only explicitly mention later in that paragraph. It is unclear to this reviewer, why the authors call it IDH1-EX527.

Thank you for pointing out the unclear description in our manuscript. We apologize for the confusion caused by the unclear statement. We have revised the manuscript accordingly. The compound EX-527 is a SIRT1 inhibitor, and we have corrected the description to simply "EX-527" in the revised manuscript.

(11) The end of the introduction needs revising to be more concise; the last paragraph on p. 4 ("Recently, the decreased expression of IDH1...") partially should be integrated with the previous paragraph, and partially is repeated in the last paragraph (top paragraph on p. 5). The last sentence on p. 4, "These findings strongly suggest that aberrant expression of IDH1 is also an important factor in the pathogenesis of AML and MDS.", should rather read "increased expression of IDH1", to distinguish it from mutant IDH1 (mutant IDH1 is also aberrantly expressed IDH1).

We appreciated the reviewer for the helpful suggestion. Considering that the inclusion of this paragraph did not provide a valuable contribution to the formulation of the scientific question, we have removed it after careful consideration, and the revised manuscript is generally more logically smooth.

(12) Abstract and last sentence of the introduction: "innovative perspective" should be re-worded, as the authors present data, not a perspective. Maybe could use "evidence".

Thanks for the reviewer’s suggestion. It has been revised accordingly.

(13) "IDH1-mut AML/MDS" on p. 11. The authors should provide more information about these AML/MDS samples. The legend contains no information about them/their mutational status. How many samples did the authors look at? Do these cells contain mutations other than IDH1?

Thanks for the reviewer’s suggestion. The detail information of these AML/MDS samples are provide in supplemental table 1. In our current study, we collected ten AML/MDS samples and the majority of the samples only contain IDH1 mutations at different sites.

(14) The statement, "Taken together, these results indicated that IDH1 deficiency reshaped chromatin states and subsequently altered gene expression pattern, especially for genes regulated by H3K79me3, which was the mechanism underlying roles of IDH1 in modulation of terminal erythropoiesis." (p. 10), is not correct at that point in the manuscript as the authors have not yet introduced the RNA-seq data.

Thanks for the reviewer’s suggestion. The statement has been revised to “Taken together, these results indicated that IDH1 deficiency reshaped chromatin states by altering the abundance and distribution of H3K79me3, which was the mechanism underlying roles of IDH1 in modulation of terminal erythropoiesis”.

(15) For easier readability, the authors should present the data in order. For example, the supplemental data for IDH shRNA and siRNA should be presented together and not in Supplementary Figures 1 and 5. Supplementary Figure 3 is mentioned after Supplementary Figure 1, but before Supplementary Figure 2 - again, all data need to be presented in subsequent figures to be viewed together.

Thank you for your suggestion regarding the order of data presentation. We have reorganized the figures in the manuscript to improve readability. We apologize for any confusion caused by the previous arrangement and hope that the revised version meets your expectations.